# A naturally occurring mitochondrial genome variant confers broad protection from infection in *Drosophila*

Tiina S. Salminen[1,2,&,‡,*], Laura Vesala[1,3,&], Yuliya Basikhina[1], Megan Kutzer[2], Tea Tuomela[1], Ryan Lucas[2], Katy Monteith[2], Arun Prakash[2], Tilman Tietz[1,3], Pedro F. Vale[2,‡]

1 Mitochondrial Immunometabolism research group, Faculty of Medicine and Health Technology, Tampere University, Tampere, Finland, 2 Institute of Ecology and Evolution, School of Biological Sciences, University of Edinburgh, Edinburgh, United Kingdom, 3 Department of Molecular Biology, Umeå University, Umeå, Sweden

& These authors contributed equally to this work.
‡ These authors are joint senior authors on this work.
* tiina.s.salminen@tuni.fi

**Data Availability Statement:** Data and code for statistical analyses are available at Zenodo (https://zenodo.org/records/14162508). The RNA sequencing data is available at NCBI's Gene

## Abstract

The role of mitochondria in immunity is increasingly recognized, but it is unclear how variation in mitochondrial DNA (mtDNA) contributes to variable infection outcomes. To quantify the effect of mtDNA variation on humoral and cell-mediated innate immune responses, we utilized a panel of fruit fly *Drosophila melanogaster* cytoplasmic hybrids (cybrids), where unique mtDNAs (mitotypes) were introgressed into a controlled isogenic nuclear background. We observed substantial heterogeneity in infection outcomes within the cybrid panel upon bacterial, viral and parasitoid infections, driven by the mitotype. One of the mitotypes, mtKSA2, protected against bacterial, parasitoid, and to a lesser extent, viral infections. Enhanced survival was not a result of improved bacterial clearance, suggesting mtKSA2 confers increased disease tolerance. Transcriptome sequencing showed that the mtKSA2 mitotype had an upregulation of genes related to mitochondrial respiration and phagocytosis in uninfected flies. Upon infection, mtKSA2 flies exhibited infection type and duration specific transcriptomic changes. Furthermore, uninfected mtKSA2 larvae showed immune activation of hemocytes (immune cells), increased hemocyte numbers and ROS production, and enhanced encapsulation response against parasitoid wasp eggs and larvae. Our results show that mtDNA variation acts as an immunomodulatory factor in both humoral and cell-mediated innate immunity and that specific mitotypes can provide broad protection against infections.

## Author summary

The strength of immune response and the disease symptoms vary among individuals even when exposed to the same pathogen. Much of this variation is due to the genes directly involved in adaptive and innate immune pathways. In addition, mitochondria are

Expression Omnibus (GEO) data repository under the accession number GSE260986.

**Funding:** Financial support was provided by the Academy of Finland (322732, 328979 and 353367) and the Sigrid Jusélius foundation (3122800849) grants to TSS, Tampere University Doctoral School funding and The Finnish Cultural Foundation grant to YB, and a Leverhulme Trust Research Project grant (RPG-2018-369) and a Branco Weiss Fellowship to PV. The funders had no role in study design, data collection and analysis, decision to publish, or preparation of the manuscript.

**Competing interests:** The authors have declared that no competing interests exist.

emerging as one of the factors modulating immune responses. Mitochondria are cellular organelles with their own genome (mtDNA) involved in many important processes, including producing ATP, the energy currency of the cells. mtDNA mutations leading to mitochondrial dysfunction are often considered harmful upon exposure to pathogens. Here, we used a fruit fly model where unique mtDNA variants are placed in a controlled nuclear genomic background to find out how mtDNA shapes infection outcomes upon bacterial, viral and parasitoid infections. We found that mtDNA variation alters the efficiency of immune response and describe a mtDNA variant that confers protection against variety of pathogens. This protection was at least partially caused by enhanced cell-mediated innate immunity, including higher numbers of immune cells prior to and during infection.

## Introduction

Immune responses among individuals vary tremendously even upon exposure to the same pathogen [1]. This variation is a result of several factors, such as the age and sex of an individual, prior pathogen exposure, and various genetic factors [2–4]. However, we do not yet fully grasp the causes and extent of the heterogeneity we observe in host immune responses. A growing body of work in the last decade established that mitochondria play a crucial role in immune function and signalling in both vertebrates and invertebrates [5–7]. Mitochondria are cellular organelles that utilize dietary energy for the formation of adenosine triphosphate (ATP) via the tricarboxylic acid cycle (TCA) and oxidative phosphorylation (OXPHOS). ATP is used to fuel cellular processes including energetically costly immune responses. At the most basic level, mitochondria orchestrate immune cell function by modulating cell metabolism. However, mitochondria also facilitate the activation of innate immune responses by forming signalling complexes. In mammalian immunity, mitochondrial antiviral-signaling protein (MAVS) located at the outer mitochondrial membrane (OMM) acts as a platform for antiviral signalling, while the NLRP3 inflammasome assembles on OMM to activate inflammatory responses [8,9]. Mitochondria also produce several mediators that regulate immune function, such as reactive oxygen species (ROS) that act as signalling molecules in immune responses as well as direct antimicrobials, and metabolites from the TCA cycle, like citrate, itaconate, and succinate, that regulate cytokine secretion and immune cell activation [5,7]. Finally, mitochondrial components can act as Damage-Associated Molecular Patterns (DAMPs) to trigger innate immune responses directly [10–12]. Given these important roles of mitochondria in immunity, it is likely that changes in mitochondrial functions could result in variation in both cellular and humoral immune responses. However, the extent to which variation in mitochondrial function between individuals may contribute to interindividual variation in the response to pathogens is currently unclear.

Mitochondrial function is affected by complex genetic factors: 1) mitochondria contain their own genome, mtDNA, that follows a maternal inheritance pattern instead of more common biparental inheritance pattern; 2) mitochondria require the coordinated expression of nuclear and mitochondrial genes; 3) mtDNA is present in a mitochondrion in multiple similar or dissimilar copies (mtDNA copy number), leading to homo- or heteroplasmy, respectively; 4) mitochondrial mutations can be inherited and thus systemic, or sporadic and tissue-specific, and 5) OXPHOS contains the multisubunit complexes I-V (cI-cV) with protein partners encoded by both the nuclear genome (nDNA) and the mtDNA (with the exception of cII which contains nuclear encoded subunits only) and some of these complexes contain

autosomal, X-linked and mtDNA-encoded subunits, such as in the case of OXPHOS complex IV [13,14]. Changes in any of these factors and the epistatic interactions between the mitochondrial and the nuclear genome can thus affect the function of mitochondria, and therefore impact the immune response to a diverse array of pathogens.

Here, we focus on how natural mtDNA variation contributes to infection outcomes. The mtDNA is circular, circa 16.5 kb in humans and it encodes 13 protein coding genes, 22 tRNA and two rRNA genes, all involved in OXPHOS. Studying the effects of mtDNA variation can be challenging, as they can be difficult to separate from those originating from the nuclear genome or from epistatic interactions between the two genomes. However, controlling the effect of the nuclear genome is feasible with the fruit fly *Drosophila melanogaster* model via generation of cytoplasmic hybrids, aka cybrid fly lines. In cybrids, specific mtDNA variants (mitotypes) are introgressed into an isogenic nDNA background by backcrossing over multiple generations (>10) [15]. The content and function of the mtDNA is highly conserved between fruit flies and humans [15–17], making it possible to model various aspects of mtDNA variation. *Drosophila* cybrid model was previously used to examine the effect of mtDNA variation, *e.g.* on mtDNA copy number, weight, developmental time [18], sleep and locomotion [19], reproduction [20], mitochondrial disease [21] and immunity [22].

To fight infections, the vertebrate immune system reacts through the innate and, subsequently, the adaptive immune responses. Invertebrates like *D. melanogaster* rely solely on evolutionarily conserved innate immunity to combat viral, bacterial, fungal and parasitic pathogens [23]. These innate immune responses include antimicrobial peptide (AMP) production, phagocytosis, melanisation reactions as well as antiviral responses involving multiple tissues like the fat body (homologous to the liver and adipose tissue in mammals), muscles, and immune cells (hemocytes) [24]. Two NF-κB signaling pathways, the Toll and Immune deficiency (Imd) pathways, are central for the *Drosophila* humoral immune responses [25–28]. Specific peptidoglycan recognition proteins (PGRPs) activate these signaling pathways when encountering l-lysine (Lys)-type peptidoglycan (PGN), leading to Toll activation (mostly Gram-positive bacteria and fungi), or diaminopimelic acid (DAP)-type PGN (mostly Gram-negative bacteria) leading to Imd activation [29,30]. RNA interference (RNAi) triggered by double stranded RNA and mediated by the RNaseIII-like enzyme Dicer and the RNA-induced silencing complex (RISC) [31] is the main antiviral response in the fly. Finally, an invasion by parasitoid wasps, which lay their eggs in *Drosophila* larvae, activates the cellular arm of the innate immune system. Plasmatocytes comprise the majority of hemocytes in *Drosophila* larvae and adults. These cells phagocytose bacteria and cellular debris and produce AMPs [32,33], although the main AMP-producing tissue is the fat body. In larvae, plasmatocytes can differentiate into immune activated hemocytes called lamellocytes [34–36] which are essential for fighting the parasitoid wasps [37]. Plasmatocytes and lamellocytes form a multilayered capsule around the parasitoid eggs and larvae, and lamellocytes, together with a third hemocyte type, the crystal cell, produce melanin to seal the capsule [38,39].

We aimed to test how naturally occurring variation in mtDNA, with a potential to impact the function of OXPHOS complexes, modifies cellular and humoral innate immune responses following bacterial, viral, or parasitoid infections. This study was motivated by the discovery of a naturally occurring mtDNA variant of *Drosophila* called mtKSA2 causing formation of melanotic nodules in uninfected animals [21]. Melanotic nodules are melanised aggregates of hemocytes [40] and are considered an indicator of an aberrantly activated cell-mediated innate immune response. mtDNA and mitochondria related nuclear genome mutations leading to mitochondrial disease can have adverse effects on the immunocompetence in humans [41,42], and it was also shown that certain mtDNA mutations can adversely affect innate immune responses in *Drosophila* [22]. Here, we aimed to determine if specific mtDNA variants, such as

mtKSA2, can be beneficial when the host is immune challenged. Our data confirms that naturally varying mitochondrial genomes can modulate host innate immune responses, and that some mitotypes may confer protection from infection.

## Results

### Mitochondrial genome variation affects fly survival when exposed to pathogenic bacteria

To study the effect of mtDNA variation on humoral innate immune responses, we utilized a panel of *D. melanogaster* cybrid lines [18] (referred to as "mitotypes") and systemically infected them with DAP-type peptidoglycan containing Gram⁻ bacteria *Providencia rettgeri* and LYS-type peptidoglycan containing Gram⁺ bacteria *Staphylococcus aureus*. In general, females tended to be more susceptible to both bacterial species than males (Fig 1A–1B'). The cybrid lines form two haplogroups based on mtDNA sequence similarity [18] and especially haplogroup I females were more susceptible to *P. rettgeri* infection than males of the same haplogroup (S1A Fig). Furthermore, mtDNA mitotypes drastically affected the survival of the hosts after *P. rettgeri* infection, ranging from 15% (mtBS1) to 71.7% (mtKSA2) in males and from 1.7% (mtBOG1) to 33.3% (mtKSA2) in females (Fig 1A-1A'). The survival rates of *S. aureus*-infected mitotypes ranged from 16.7% (mtM2) to 59.3% (mtKSA2) in males and from 5% (mtBS1 and mtM2) to 63.3% (mtKSA2) in females (Fig 1B-1B'). mtDNA haplogroup -specific sexual dimorphism was also apparent in *S. aureus*-infected flies (S1B Fig). In general, mtKSA2 females and males had the highest survival rates in both infections when compared to the other mitotypes (S2 Fig).

### mtKSA2 induced increase in survival is due to tolerance rather than increased bacterial clearance

Host disease defense occurs through a combination of resistance and tolerance. Resistant flies can clear an infection, whereas tolerant flies are able to limit the infection-induced damage without clearing the pathogen [43,44]. To further study the effect of mtDNA mitotypes on infection outcomes, we measured bacterial loads (colony forming units, CFUs) in four mitotypes selected based on their survival rates after exposure to pathogenic bacteria (Fig 1A-1B') to represent high (mtKSA2 and mtWT5A) and low (mtORT and mtBS1) survival rate lines (S2 Fig). We measured CFUs at an early-stage infection time point (8 hours post infection, h *p. i.*) and when the flies began to die (20 h *p.i.*) after exposure to *P. rettgeri* or *S. aureus*. We additionally measured CFUs at late-stage infection time points, *i.e.* when survival rates started to plateau, at 48 hours after *P. rettgeri* and 72 hours after *S. aureus* infection. *P. rettgeri* load differed among mitotypes at 8 h *p.i.* (Mitotype: p = 0.049) and there was a differential effect of mitotype and sex at 20 h *p.i.* (Mitotype x Sex: p = 0.03). *S. aureus* load differed between males and females (Sex: p = 0.0003) at 8 h *p.i.* and there was a differential effect of mitotype and sex at 20 h *p.i.* (Mitotype x Sex: p = 0.0006) and 72 h *p.i.* (p = 0.004) (Fig 1C-1D'). However, this did not directly reflect the differences seen in the survival rates. Therefore, we plotted the average CFUs quantified during late-stage infections against survival rates to better visualize average disease susceptibility among mitotypes [45,46] (Fig 1E-1F'). We found that mtKSA2 males infected with *P. rettgeri* appeared to be more tolerant because they had high average pathogen burden and higher survival compared to the other mitotypes (Fig 1E). Females in each of the four mitotypes had similar *P. rettgeri* burdens but varying survival rates, which is suggestive of variation in disease tolerance. mtORT and mtBS1 females appeared to be more susceptible to *P. rettgeri* infection compared with mtKSA2 and mtWT5A, illustrated by their low survival

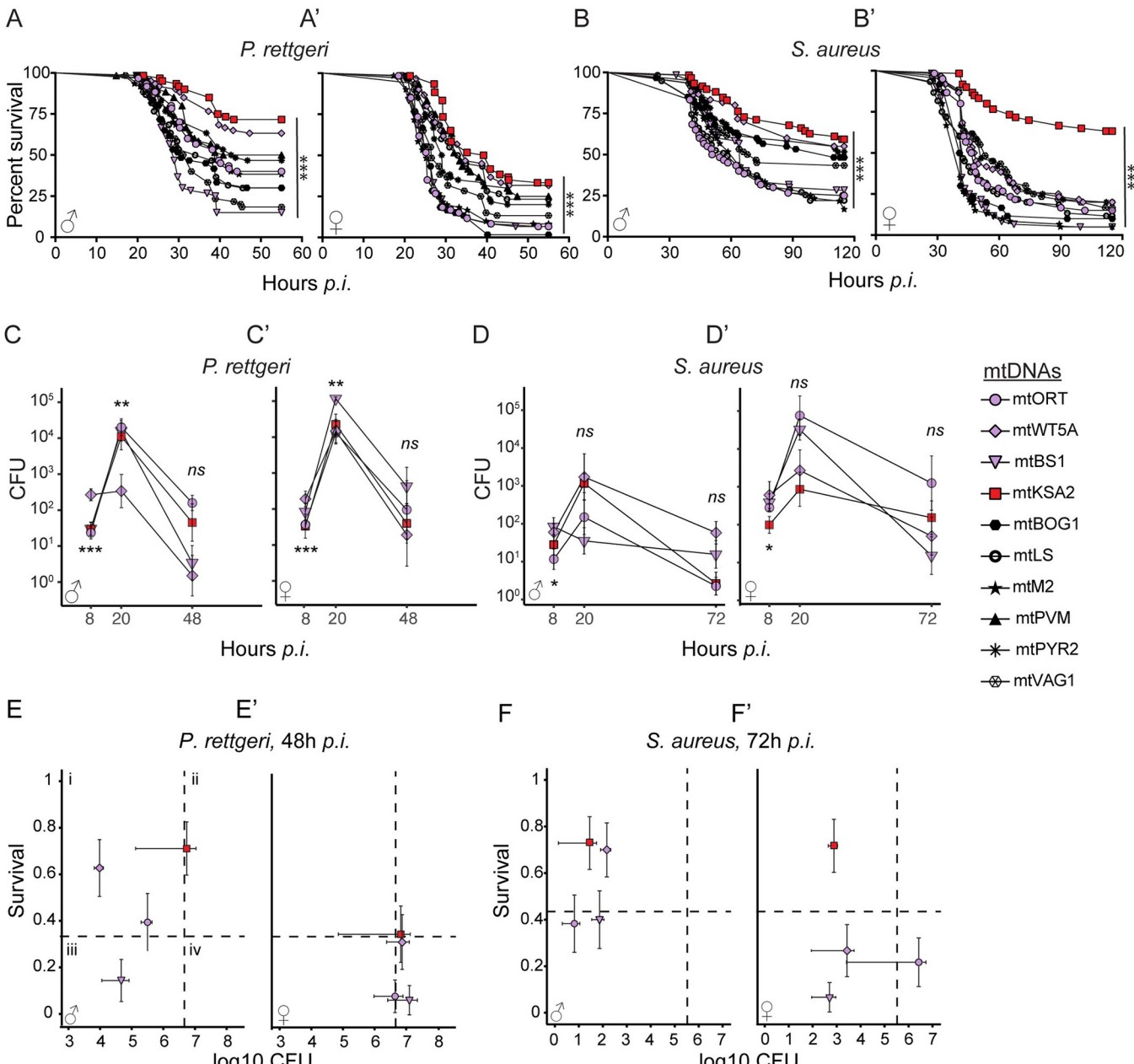

**Fig 1. mtDNA variation affects infection outcomes after exposure to bacterial pathogens.** (**A-A'**) Males and females of nine cybrid lines and nuclear background line Oregon RT (ORT) were infected with Gram-negative *P. rettgeri* (n = 60) or with (**B-B'**) Gram-positive *S. aureus* bacteria (n = 60) (Note that mtPVM is missing from the *S. aureus* infection). Stars denote the overall statistical difference in the survival among all the mitotypes analysed using the Mantel-Cox log rank test. The statistical differences in the survival rate of the survival-enhancing mitotype mtKSA2 (red) in comparison to other mitotypes are shown in S2 Fig Mitotypes mtWT5A (high survival rate; purple), mtBS1 and mtORT (low survival rate; purple) were studied further alongside mtKSA2. (**C-D'**) Bacterial loads were measured as colony forming units (CFUs) 8 and 20 hours post infection (h *p.i.*) and in addition 48 h after infection with *P. rettgeri* and 72 h after *S. aureus* (n = 8). The data on CFUs was analysed with linear models with Type III Sums of Squares. (**E-F'**) Resistance vs. tolerance quadrants. We plotted mean (**E-E'**) *P. rettgeri* and (**F-F'**) *S. aureus* CFUs measured at 48 and 72 h *p.i.* against the proportion of surviving flies to summarize potential differences between resistance and tolerance among four mitotypes, focusing on mtKSA2. Quadrants (dashed lines) represent the cut-off points between different defence phenotypes determined by overall mean survival and colony forming units: **i**) resistant (healthy), **ii**) tolerant (healthy), **iii**) resistant (pathology) and **iv**) failing. *ns*, not significant; \*, p<0.05; \*\* p<0.01; \*\*\*, p<0.001.

rates (Fig 1E'). In contrast, *S. aureus*-infected males tended to carry similar bacterial burdens but survival rates among mitotypes differed. mtKSA2 and mtWT5A males had higher survival rates than mtBS1 and mtORT males, suggesting that mtKSA2 and mtWT5A were more tolerant (Fig 1F). In *S. aureus* infected females, mtKSA2 had the highest survival rate with bacterial load similar to mtWT5A and mtBS1, while mtORT had both high CFUs and low survival (Fig 1F'). Taken together, the higher survival rates of mtKSA2 were generally due to increased tolerance, and not to a greater ability to clear infection.

## Immune priming is mitotype-specific

To test if the increased survival of mtKSA2 upon bacterial infection could be caused by preactivation of innate immune responses we primed female and male flies with heat inactivated *P. rettgeri* and challenged them with live *P. rettgeri* 18 hours later. Our expectation was that if mtKSA2 possesses preactivated (humoral) immune response prior to infection, the line should have a weaker response to priming. We found that the effect of priming varied among the mitotypes (Mitotype x Priming: p = 0.002) (S2 Fig). Survival of mtWT5A and mtBS1 flies was higher after being primed with heat inactivated bacteria and challenged with live bacteria (S3 Fig). Conversely, mtKSA2 survival and mtORT male survival following *P. rettgeri* infection was not significantly affected by priming. Therefore, the protective effect of immune priming seems to depend on mitotype, and mtKSA2 flies did not respond to priming based on their survival rates.

## mtKSA2 antiviral protection is less pronounced than antibacterial protection

The effect of mtDNA variation was examined upon infection with two *Drosophila*-specific viruses, Kallithea Virus (KV; [47]) and Drosophila C Virus (DCV; [48]). Kallithea virus (family Nudiviridae) is a large double stranded DNA virus and DCV (family Dicistroviridae) is a single stranded positive sense RNA virus. Because KV does not affect female survival [47], we only infected males and followed their survival rates for 22 days. mtORT, mtWT5A and mtBS1 showed almost identical end-point survival rates (7–14%) upon KV infection, while mtKSA2 again showed the best survival rate among the studied mitotypes, as 38% of mtKSA2 males survived the infection (Figs 2A and S4A). We followed male and female survival for 14 days after DCV infection. During the acute phase of the DCV infection, mtORT males (Fig 2B) and females (Fig 2B') died at a faster rate than the other mitotypes, but the end-point survival percentage did not differ. However, overall mtORT was shown to be more susceptible to DCV infection than mtKSA2 (S4B-S4B' Fig).

## mtDNA variation affects the transcriptome profiles of uninfected flies

We used RNA sequencing to examine the effect of mtDNA variation on the nuclear and mitochondrial transcriptomes. Bulk RNA sequencing was performed on uninfected mtKSA2 and mtORT flies as well as on the same fly lines infected with *P. rettgeri*, *S. aureus* (8 and 20 h *p.i.*) or DCV (3 d *p.i.*). Based on a principal component analysis (PCA) of the normalized read counts, female and male samples formed their own markedly distinct clusters (Fig 3A, sex explaining 82% of the variation in the data). In males, mtKSA2 and mtORT samples were separated from each other by variable degrees depending on infection type and timepoint (Fig 3A'–3A'''). In uninfected and in *P. rettgeri*-infected females, mitotypes showed separate clustering (S5A Fig) whereas in *S. aureus* and DCV infected females clustering according to mitotype and infection timepoint was less obvious (S5A'–S5A" Fig). Overall, the PCA indicates that the two mitotypes cause variation in the transcriptomes of the uninfected flies, as well as upon

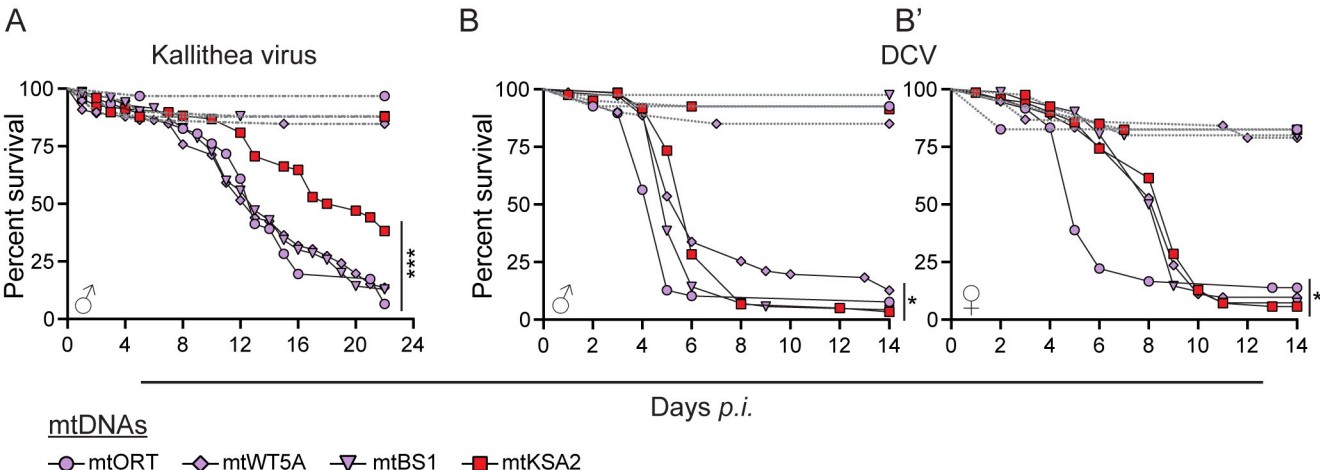

**Fig 2. mtKSA2 effect upon viral infection is not as strong as upon bacterial infection.** (**A**) Survival rates of males from the four mitotype lines after infection with double stranded DNA virus Kallithea. Controls for Kallithea virus infection were injected with 1 x PBS (dashed gray lines). (**B-B'**) Survival of both males and females after infection with single stranded positive sense RNA virus Drosophila C virus (DCV). Controls for DCV infection were pricked with Tris-Hcl solution (dashed gray lines). Overall differences in the survival among the mitotypes were analysed using the Mantel-Cox log-rank test. Differences in comparison to survival-enhancing mitotype mtKSA2 (red) are shown in S4 Fig. (**A-B'**) *ns* not significant; * p<0.05; ** p<0.01; *** p<0.001.

different infections. To emphasize the effects of mtDNA mitotypes and infection on the transcriptomes, we focused on the male transcriptome data in the present work and not on sex specific differences. The female data is available under the same GEO access number as the male data (GSE260986). The data of male log2FCs and false discovery rate (FDR) adjusted p-values are also listed in S1 Table.

We utilized Gene Ontology (GO) enrichment analyses (https://www.flymine.org/flymine; [49]) to compare the differentially expressed (DE) genes between uninfected mtKSA2 and mtORT males. Based on the enriched biological processes, a large set of the upregulated genes in mtKSA2 belonged to terms related to energy generation via mitochondrial respiration (S2 Table and Fig 3B). Other representative GO terms were associated with "cuticle development", "muscle system process" and "glycolytic process" (S2 Table and Fig 3B). Among the genes downregulated in mtKSA2, "response to bacterium" (23 genes) was the only significantly enriched GO term (Fig 3B) which was surprising given mtKSA2 flies' enhanced survival rates upon bacterial infections (Fig 1). Seven of these genes encode proteins with antimicrobial function (*Diptericin A*, *Diptericin B*, *Defencin*, *Drosocin*, *Attacin-A*, *Bomanin Bicipital 1* and *Baramycin*), three encode for *negative* regulators of the Imd pathway (*pirk*, *PGRP- SC2* and *Charon*) and two genes are related to stress responses (*Turandot E* and *p38c MAP kinase*). However, several of these genes are not directly connected to the core immune response (such as four genes encoding for lysozymes and a gene encoding for a myosin ATPase; S2 Table). Hence, we next examined the expression of a larger set of AMPs as well as key components of Toll and Imd pathways [26–28] in uninfected mtKSA2 and mtORT flies (Fig 3C and S3 Table). Besides the AMPs included in the significantly downregulated GO term "response to bacterium", most of the other AMPs were either downregulated or not significantly affected in uninfected mtKSA2 when compared to mtORT (Fig 3C and S3 Table). Only *Drosomycin*, *Metchnikowin-like* and *Attacin-C* were significantly upregulated. Furthermore, Toll and Imd pathway components were mostly unaffected by mtDNA mitotype in the absence of infection, except for the negative regulators of the Imd pathway mentioned above. Of note, both bacterial infections and DCV infection caused a robust and consistent upregulation of the AMPs (S3 Table). This response was similar in mtKSA2 and mtORT. Likewise, *P. rettgeri* and *S.*

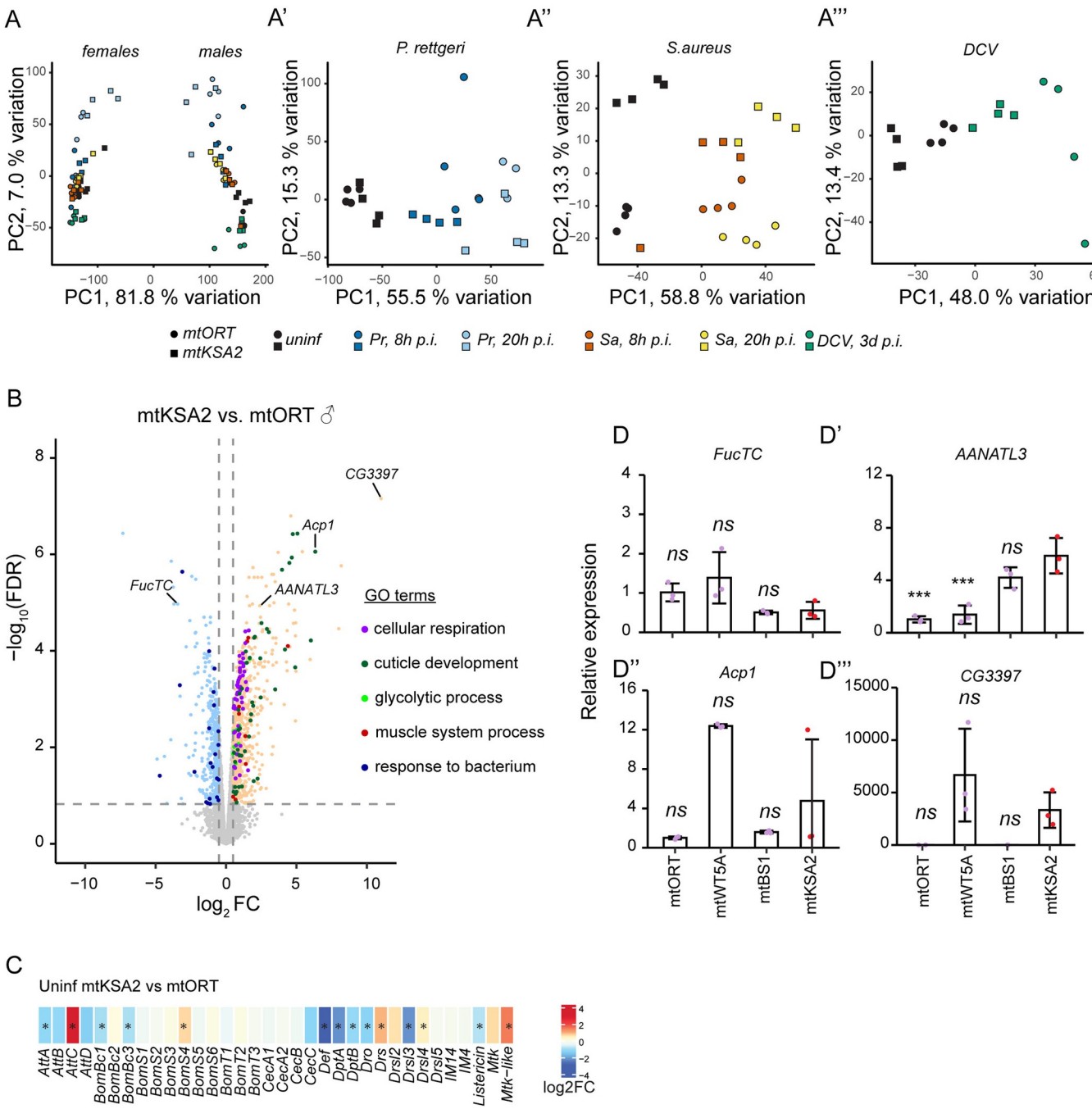

**Fig 3. Overview of RNA sequencing sample clustering and differentially expressed genes.** (**A'-A'''**) Principal component analysis (PCA) plots of (**A**) all RNA sequencing samples, females (left) and male (right) samples showing a clear sex-specific clustering pattern, (**A'**) *P. rettgeri* infected and control, (**A''**) *S. aureus* and control and (**A'''**) DCV infected and control mtORT and mtKSA2 cybrid male samples. Each treatment includes four biological replicates containing pools of 10 individuals. (**B**) Volcano plot showing differentially expressed genes in uninfected mtKSA2 vs. mtORT males. Log₂FC cut-off was set to 0.5 and FDR-adjusted p-value to 0.15. Genes in the main GO terms are shown with different colors and genes tested with RT-qPCR are labelled in the volcano plot. (**C**) The expression levels of common Toll and Imd pathway target AMPs in uninfected mtKSA2 and mtORT. Expression of these AMPs upon *P. rettgeri*, *S. aureus* and DCV infections is listed in S3 Table. Stars denote statistically significant fold changes (FDR< 0.15). (**D-D'''**) Expression levels of *FucTC*, *AANATL3*, *Acp1* and *CG3397* were measured with RT-qPCR in uninfected males of the four cybrids lines (n = 3, pools of 5). Data was analysed using one-way analysis of variance followed by Tukey's HSD pairwise comparisons. Differences to mtKSA2 are indicated. *ns* not significant; * p<0.05; ** p<0.01; *** p<0.001.

*aureus* infections led to the upregulation of the upstream activators of Toll and *Toll* itself in mtKSA2 and mtORT. The Imd-activating *PGRP-LC* was upregulated in both mitotypes after infection by *P. rettgeri*, but not after *S. aureus* (except for a mild upregulation in mtORT 8 h *p. i.*), and *Imd* was upregulated by both *P. rettgeri* and *S. aureus* at 8 h *p.i.* Taken together, although uninfected mtKSA2 showed suppression of many genes related to NF-κB pathways, they were robustly and similarly induced in mtKSA2 and mtORT flies upon infection.

When comparing the transcriptomes of two mitotypes, one cannot conclude if the genes of interest are specifically up- or downregulated in the mitotype of interest. Hence, we measured the expression patterns of the representative genes with RT-qPCR from uninfected mtORT, mtKSA2, mtBS1 and mtWT5A flies, with a new set of male samples. We picked the gene with the highest fold change (FC) in uninfected mtKSA2 when compared to mtORT (CG3397); a highly expressed gene representing the GO term cuticle development (Acp1; Adult cuticle protein 1); and two genes that were either downregulated (FucTC; alpha1,3-fucosyltransferase C) or upregulated (AANATL3; Arylalkylamine N-acetyltransferase-like 3) in uninfected mtKSA2 in comparison to mtORT flies (Fig 3B). None of the genes showed consistent, mtKSA2 specific expression pattern, but instead FucTC (Fig 3D) and AANATL3 (Fig 3D') were similarly expressed in mtORT and mtWT5A (highest mtDNA sequence similarity [18]), when compared to mtKSA2 and mtBS1, whereas Acp1 (Fig 3D") and CG3397 (Fig 3D''') had similar expression patterns in mtORT and mtBS1 (the two susceptible lines) when compared to mtWT5A and mtKSA2.

## Mitotypes shape the transcriptome profiles upon infection

We performed a GO term analysis to obtain an overview of the transcriptomic profiles in mtKSA2 and mtORT when exposed to pathogens. The GO analysis showed that mtORT and mtKSA2 flies had overlapping responses to *P. rettgeri* and *S. aureus*, but there also was a large set of differentially expressed genes that were mitotype-specific (Fig 4A-4D'). Furthermore, more genes were differentially expressed at the early *p.i.* timepoint than later, and the enriched GO terms varied between timepoints, likely reflecting differences in the initiation phase of the immune response at 8 h *p.i.* and later at the 20 h *p.i.* Overall, the GO term analysis yielded numerous significant GO terms, full lists of which are presented in S4 Table and a manually curated overview in Fig 4.

Genes upregulated in both mitotypes at 8 h and 20 h after *P. rettgeri* and *S. aureus* infection were involved in such processes as protein synthesis and immune and stress responses (Fig 4A-4B and 4C-4D). Shared downregulated genes were more variable and were involved in GO terms "cilium organization", "cytoskeleton organization", "protein polyglycylation" and "mitochondrial transport", depending on infection type and timepoint (Fig 4A'–4B' and 4C' and 4D'). In addition, multiple developmental processes were downregulated specifically at 8 hours post *P. rettgeri* infection (S4 Table).

Genes related to GO terms "cell cycle" and "cell differentiation" were upregulated in mtKSA2 after *P. rettgeri* infection (Fig 4A and 4C). After *S. aureus* infection, mtKSA2-specific terms were largely related to development, including tissue and organ development (Fig 4C and 4D). Downregulated mtKSA2-related GO terms were involved in "cilium organization" and "cilium assembly" and "cuticle development" (Fig 4A'–4B' and 4C'–4D'). Interestingly, at 20 h post *P. rettgeri* infection, terms related to "mitochondrial transport" and "protein localization to mitochondria" were downregulated in mtKSA2 (Fig 4B' and S4 Table). In contrast, in mtORT mitochondria-related GO terms such as "cellular respiration" and "mitochondrial translation" were upregulated at 20 h after both infection types (Fig 4B and 4D). Many GO terms related to development were downregulated in mtORT (Fig 4A'–4B' and 4C'–4D').

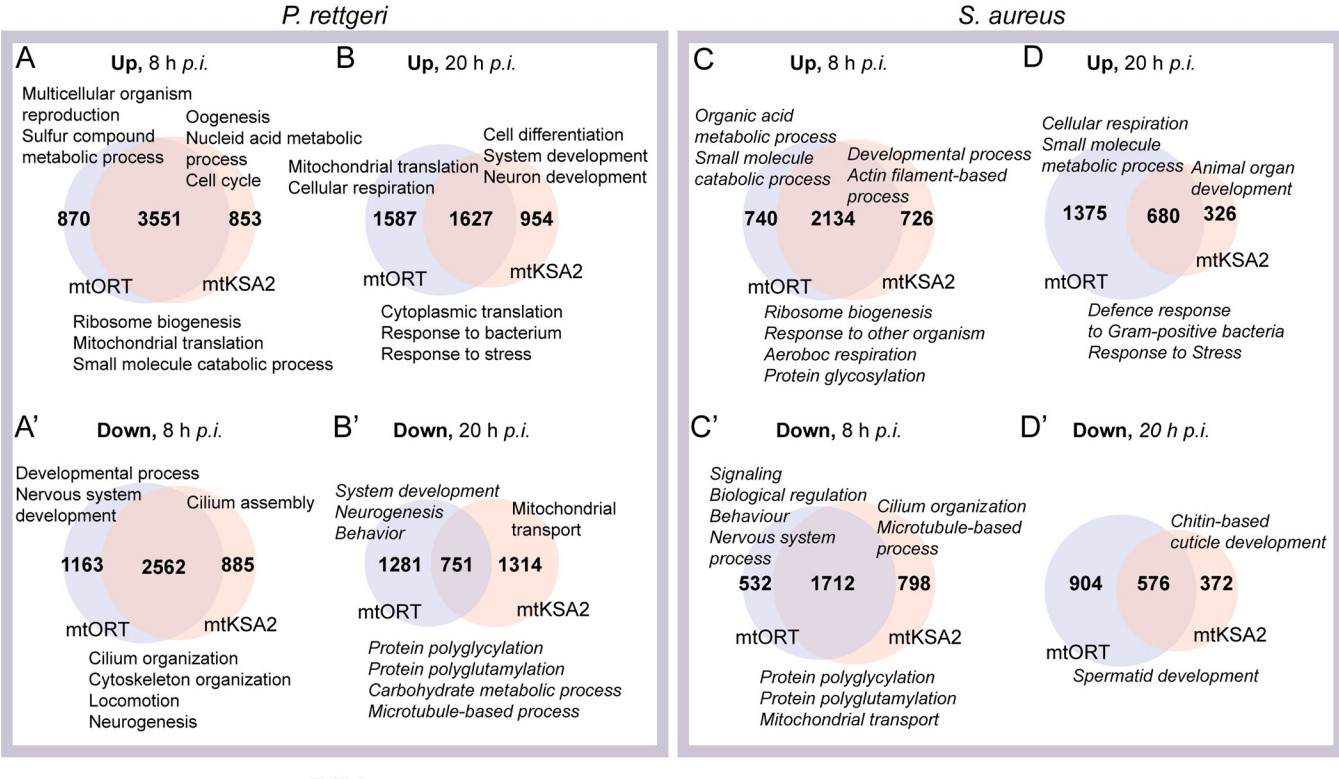

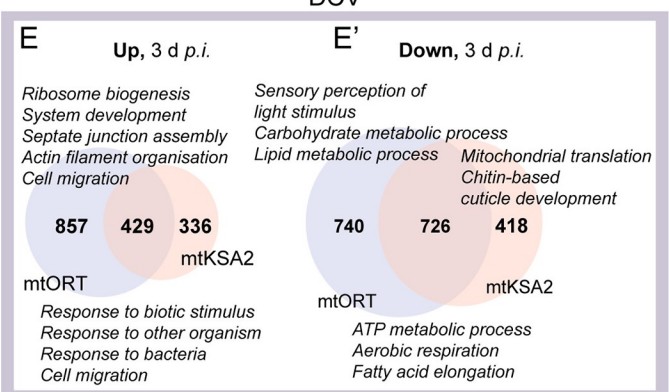

**Fig 4. Gene Ontology (GO) term analysis.** (**A-B'**) Number of genes and representative enriched GO terms (Biological process) up or downregulated after *P. rettgeri* infection in both mitotypes (intersection), in mtORT only (left) and in mtKSA2 only (right). (**C-D'**) Number of genes and representative GO terms up or downregulated after *S. aureus* infection and (**E-E'**) number of genes and representative GO terms up or downregulated after DCV infection. GO terms listed below the Venn diagrams denote the shared processes, terms listed above on the left denote processes specific to mtORT and on the right those specific to mtKSA2. See S4 Table for a full list of the significant GO terms in each treatment.

mtKSA2 and mtORT were equally susceptible to DCV infection at the end-point survival (Fig 2B) but had surprisingly low overlap in differentially regulated genes after DCV infection when compared to their uninfected state, especially when looking at the upregulated genes (Fig 4E-4E'). Among the shared, significantly enriched upregulated processes were immune response related terms such as "response to other organisms" (Fig 4E). Genes related to mitochondrial processes such as "ATP production" and "aerobic respiration" were downregulated after DCV infection in both mitotypes (Fig 4E'). mtKSA2-specific upregulated genes did not form any significant GO terms, whereas mtORT showed upregulation of "ribosome biogenesis", "cell migration" and multiple terms related to development (Fig 4E; S4 Table). mtKSA2

flies further downregulated genes related to mitochondrial function as well as cuticle development, and mtORT showed downregulation of several metabolic processes including carbohydrate, fatty acid and oxoacid metabolism (Fig 4E' and S4 Table)

Overall, bacterial and viral infections induced varied responses at the gene expression level in the mtDNA mitotypes with the core immune-responsive genes equally affected in both mitotypes. Furthermore, mitochondrial processes seem to be upregulated in both mitotypes early after bacterial infection, but only in mtORT flies at the later timepoint, whereas DCV infection caused downregulation of various mitochondria related processes in both mitotypes.

## OXPHOS gene expression is affected by mitotype and infection

Genes related to respiration were upregulated in uninfected mtKSA2 flies (Fig 3B), and genes belonging to multiple mitochondria-related GO terms were affected after bacterial and viral infections (Fig 4). We therefore delved deeper into the expression patterns of OXPHOS genes in uninfected and infected flies. A number of genes classified in the *Drosophila* database FlyBase (https://flybase.org; [50]) as OXPHOS genes, with an addition "-like (L)" in the gene name (S6 Fig), did not respond to infections in the same way as the majority of the OXPHOS genes (S6 Fig). These genes are categorized as OXPHOS genes based only on sequence similarity. They have not been experimentally linked to OXPHOS, nor do they have mammalian homologues [50]. Based on limited data availability, we speculate that they could be pseudogenes and possibly not encoding OXPHOS proteins.

We found that uninfected mtKSA2 had consistently elevated gene expression of both nuclear and mitochondrial encoded subunits of the OXPHOS complexes (Fig 5A). Next, the OXPHOS gene expression patterns were followed in mtKSA2 and mtORT flies after infection compared to their respective uninfected states. At 8 h *p.i.*, both mitotypes showed upregulation of most OXPHOS genes, with stronger upregulation in response to *P. rettgeri* infection. In general, OXPHOS gene expression was higher at 8 h *p.i.* in mtKSA2. At 20 h *p.i.*, mtKSA2 flies did not have upregulated OXPHOS gene expression to the same extent as mtORT, possibly because most of the mtKSA2 flies were already recovering from the infection and thus mitochondrial metabolism was returning to the baseline level. In contrast, upon DCV infection, most OXPHOS subunit genes were downregulated in both mitotypes, indicating a specific mitochondrial response induced by a viral infection.

Because OXPHOS gene expression differed between mitotypes, infections and time points, we examined whether mtDNA copy number varied among these treatments. For this, we collected mtORT and mtKSA2 males at 8 and 20 h *p.i.*, as well as uninfected controls. However, we detected no differences in the mtDNA copy numbers after *P. rettgeri* (Fig 4B) or after *S. aureus* (Fig 4B') infections between the two mitotypes. To investigate if the elevated expression of OXPHOS genes in mtKSA2 was reflected in the performance of mitochondria, we measured oxygen consumption (nmol $O_2$/ min/ml) of OXPHOS complexes cI, cIII and cIV in uninfected males of the four mitotypes, and did not detect significant differences in respiration between the mitotypes (Fig 4C-4C"). In conclusion, there are distinct changes in the OXPHOS gene expression patterns between mtKSA2 and mtORT flies and in how these two mitotypes respond to bacterial infection. However, these expression differences are not reflected in mtDNA copy number or respiration, and therefore the elevated OXPHOS gene expression might represent a compensatory response in mtKSA2.

## ROS-related gene expression is affected by mitotype and infection, but hydrogen peroxide levels do not vary among the mitotypes

One of mitochondrial contributions to the host response against infections is the generation of ROS, mainly via OXPHOS complexes I and III [51]. mtKSA2 flies possess the OXPHOS

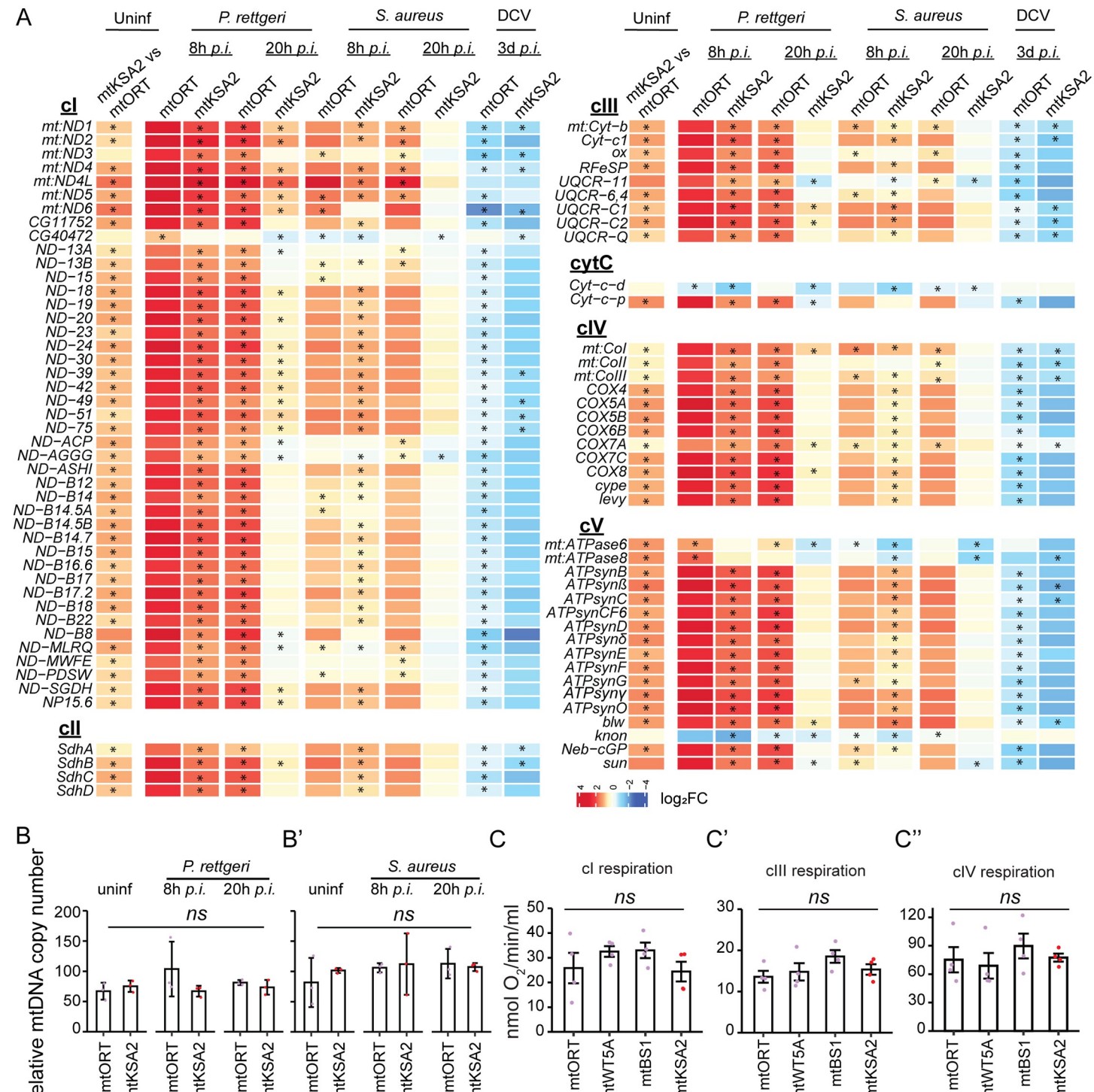

**Fig 5. OXPHOS gene expression is modulated by mtDNA and the infection type.** (A) The expression levels of OXPHOS complex genes. Stars denote statistically significant fold changes (FDR<0.15). (B-B') Mitochondrial copy number was measured as copies of the mtDNA target gene *16S* relative to nuclear target gene *RpL32*. Copy number was measured in uninfected and bacterial infected cybrid males (n = 3, pools of 5). (C-C'') Mitochondrial respiration was measured in uninfected males by measuring oxygen consumption of the OXPHOS complexes I, III and IV (n = 4, pools of 200). Data in C & D was analysed using one-way analysis of variance. *ns* not significant; * p<0.05; ** p<0.01; *** p<0.001.

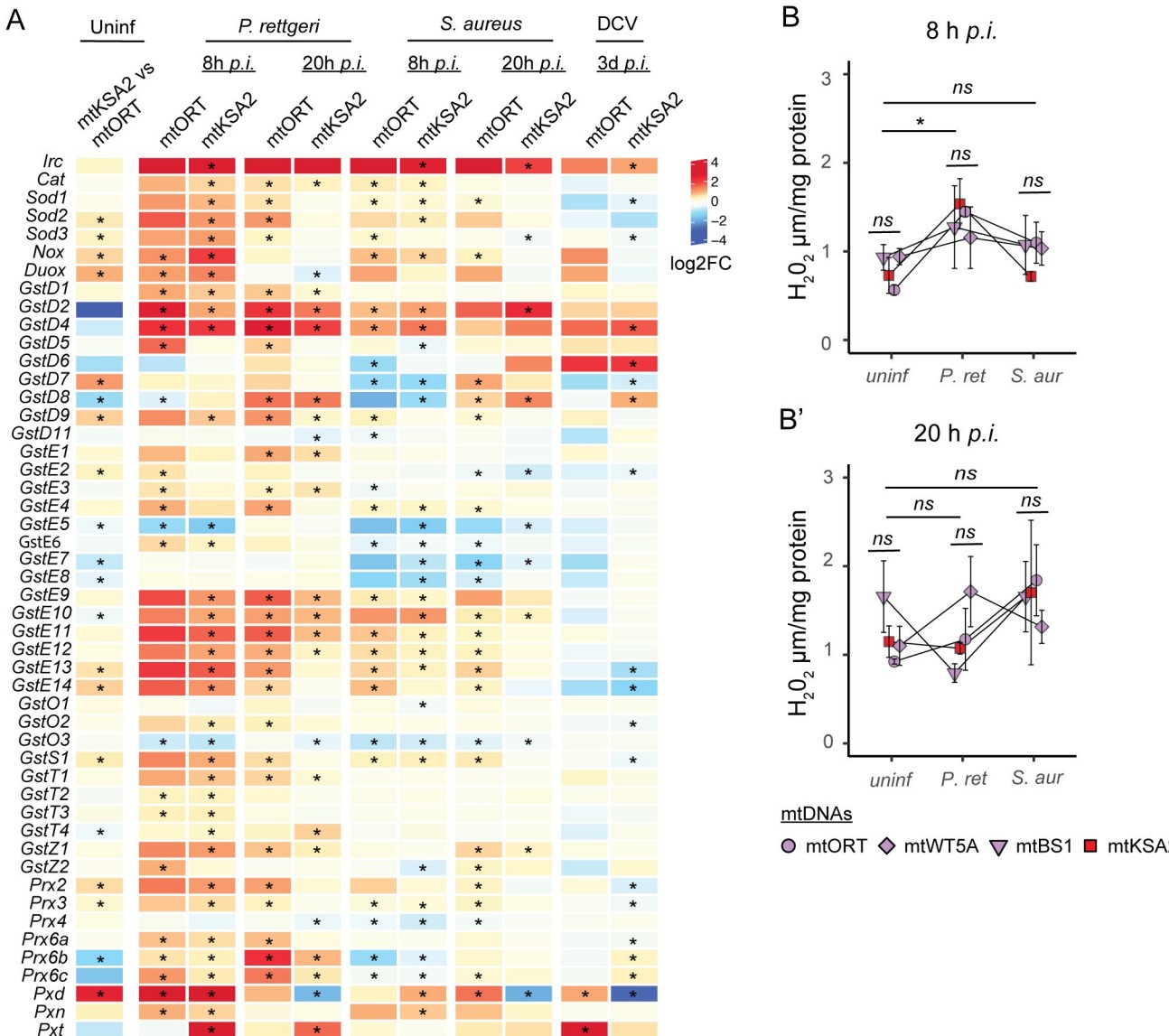

**Fig 6. Expression of ROS scavenging and producing genes and hydrogen peroxide levels vary depending on mtDNA mitotype and infection type.**
(**A**) Log2 fold changes of ROS-related gene expression in uninfected mtKSA2 flies compared to uninfected mtORT and upon bacterial and viral infections compared to uninfected flies of the same mitotype. Stars denote statistically significant fold changes (FDR< 0.15). (**B-B'**) Hydrogen peroxide ($H_2O_2$) measured from uninfected and infected male flies 8h (**B**) and 20h (**B'**) post infection. Note that the 8h and 20h timepoints were separate experiments and therefore not directly comparable to each other. Data were analyzed using linear models with a natural log transformed response where appropriate (n = 3, pools of 5 flies). Although we observed no difference among mitotypes, $H_2O_2$ significantly increased at 8 h *p.i.* with *P. rettgeri* (p = 0.0127). *ns* not significant at the level of p>0.05; * p<0.05. Error bars represent +/- one standard error of the mean.

complex III gene *mt:cyt-b* variant (S5 Table). Therefore, it is possible that this mitotype may affect ROS production. We compiled a list of the most common ROS-related genes, such as ROS scavenging, producing and detoxifying enzymes, and checked their expression patterns from the transcriptome data in uninfected and infected mtKSA2 and mtORT flies (Fig 6A). Mitochondrial and extracellular *Superoxide dismutases* (*Sod2* and *Sod3*, respectively) that convert superoxide to hydrogen peroxide ($H_2O_2$) were upregulated in uninfected mtKSA2. In addition, *Pxd*, a peroxidase functioning in neutralizing $H_2O_2$ was highly upregulated in mtKSA2, whereas peroxiredoxins with the same function were not consistently up- or

downregulated. There were less consistent patterns observed among the Glutathione S- trans-ferases (Gsts), functioning in a variety of detoxifying reactions (Fig 6A). Interestingly, the ROS-producing NADPH oxidases, *Nox* and *Dual oxidase* (*Duox*) were upregulated in unin-fected mtKSA2 flies.

In general, at 8 hours post *P. rettgeri* infection, we observed that many ROS-related genes were upregulated in both mitotypes. While the expression of these genes started to resemble the baseline level expression in mtKSA2 at 20 h *p.i.*, they remained upregulated in mtORT (Fig 6A). After *S. aureus* infection, ROS-related genes were upregulated to a lesser extent than after infecting with *P. rettgeri* in both mtORT and mtKSA2, and they were mostly unaffected or downregulated upon DCV infection (Fig 6A).

Heme peroxidases exploit the reduction of $H_2O_2$ to catalyze various oxidative reactions. Out of the ten *Drosophila* heme peroxidases, *Duox*, *Immune-regulated catalase* (*Irc*), *Peroxidase* (*Pxd*), *Peroxidasin* (*Pxn*) and *Peroxinectin-like* (*Pxt*) were upregulated in uninfected and/ or infected mtKSA2 (Fig 6A). As the upregulation of ROS producing and scavenging enzymes such as heme peroxidases might indicate an increase in the levels of ROS, we studied the effect of mitotypes and bacterial infection on the $H_2O_2$ levels in males from the four mitotypes. In general, $H_2O_2$ levels increased at 8 hours post *P. rettgeri* infection (Fig 6B), but although we observed increased expression of some of the ROS-related genes in mtKSA2 when compared to mtORT, this was not reflected in $H_2O_2$ levels in uninfected nor infected flies (Fig 6B-6B'). Taken together, both mitotypes induced expression of ROS genes as well as increased $H_2O_2$ after *P. rettgeri* infection, whereas there were less pronounced changes in gene expression and $H_2O_2$ level upon *S. aureus* infection. This suggests that ROS-related gene expression, in this context, can be used as a proxy of the degree of ROS present in the organisms. Based on these results, it is unlikely that changes in ROS production at the whole organism level are responsi-ble for the enhanced immunocompetence in mtKSA2 flies.

## mtKSA2 shows signs of an enhanced cell-mediated immune response based on transcriptome profile

We did not detect changes in the AMP expression levels that would explain the enhanced sur-vival of mtKSA2 flies upon bacterial infections (Fig 3C and S3 Table). However, a previous study demonstrated that mtKSA2 caused the formation of melanotic nodules when intro-gressed into different nuclear backgrounds, in the absence of infection [21]. Melanotic nodules are hemocyte aggregates and considered one of the hallmarks of activated cell-mediated innate immunity in *Drosophila* [40]. Hence, it is plausible that mtKSA2 may have more plasmatocytes or hemocyte immune activation prior to infection, explaining the appearance of melanotic nodules in uninfected larvae. Phagocytosis is an ancient cell-mediated immune defense [52] and therefore, we looked for changes in the expression of scavenger receptors expressed in hemocytes (plasmatocytes), the professional phagocytes in *Drosophila*, in our whole fly RNA sequencing samples. We found that *eater*, *Nimrod C1* (*NimC1)* and *Scavenger receptor class C*, *type I* (*Sr-CI*) were elevated in uninfected mtKSA2, and also following bacterial infection at some of the timepoints (Fig 7 and S6 Table). In addition to these well-characterized receptors which aid in bacterial clearance, *simu* (*NimC4;* functioning in clearance of apoptotic bodies) and some of the less well characterized Nimrod-type receptors were also elevated in both unin-fected and infected mtKSA2 (Fig 7 and S6 Table). In addition, we noted increased *Hemolectin* (*Hml*) expression, a gene expressed specifically in hemocytes, in uninfected mtKSA2 flies, as well as at 8 h after *P. rettgeri* infection in both mitotypes. These results indicate that mtKSA2 flies either have more immune cells or they exhibit increased phagocytic receptor expression to facilitate phagocytosis.

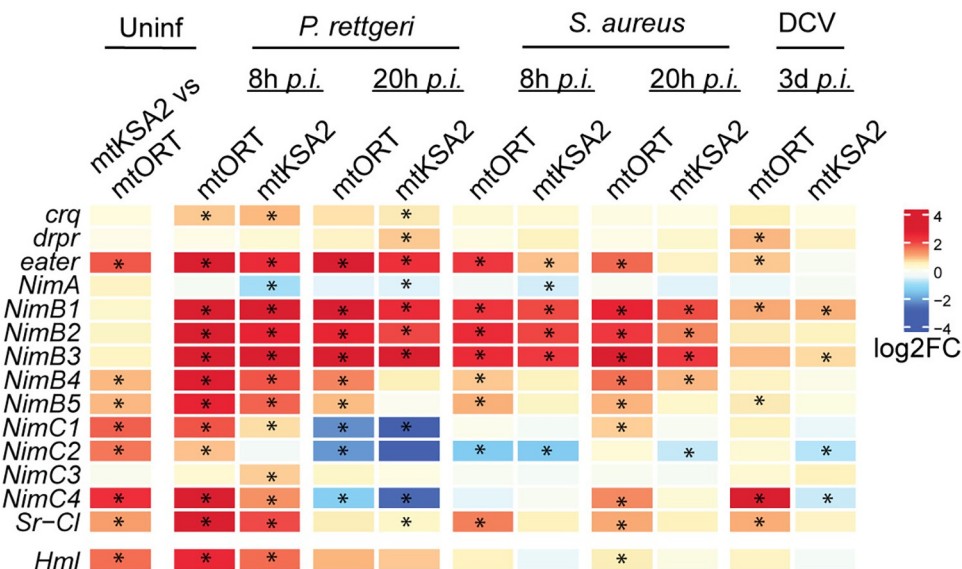

**Fig 7. Scavenger receptors expressed in hemocytes.** Expression patterns of scavenger receptors that are expressed in plasmatocytes in the uninfected mtKSA2 compared to uninfected mtORT flies, and upon different infections (mitotype compared to its uninfected basal level). Stars denote statistically significant fold changes (FDR< 0.15).

## mtKSA2 enhances the immune response after parasitoid wasp *Leptopilina boulardi* infestation

mtDNA variation, and especially mtKSA2 affected the survival rates of the flies after exposure to bacterial and viral pathogens (Figs 1 and 2). However, this was not reflected in the Toll and Imd mediated expression levels of AMPs, but instead there was evidence for increased hemocyte number in the mtKSA2 flies prior to infection (Fig 7). Furthermore, previous findings show that mtKSA2 flies exhibit melanotic nodules, a possible indication of aberrantly activated cell-mediated innate immunity [21]. Therefore, we studied the effect of mtDNA variation on the cell-mediated immune response at the larval stage. Parasitoid wasps infect *Drosophila* larvae and elicit a cell-mediated immune response. Melanotic nodules are reminiscent of the melanised capsule that is formed around parasitoid wasp eggs and larvae as a cellular defense mechanism to kill the intruder. We used the parasitoid wasp, *Leptopilina boulardi* to infect larvae of the four mitotypes and scored the efficiency of the immune response as a proportion of melanised wasp eggs and larvae found inside the fly larvae. Two rearing temperatures were used: 25˚C (bacterial and viral infections were performed at this temperature) and 29˚C (this temperature is commonly used for the wasp assays and when using the binary GAL4/UAS system for genetic manipulation [53]). In general, we found that fully (+) melanised wasp larvae were scarce in all four strains when reared at 25˚C, ranging from 0% in mtORT and mtWT5A to 5% in mtBS1 and 9% in mtKSA2 (Fig 8A). Larvae reared at 29˚C displayed an enhanced encapsulation response and mtKSA2 larvae were able to melanise (both partially and fully) more wasp larvae than the other cybrids (Fig 8A), showing that mtKSA2 enhances the cell-mediated innate immune response at the larval stage.

## mtKSA2 larvae have higher hemocyte counts pre and post infection and a stronger melanisation response

Hemocyte proliferation and lamellocyte differentiation are necessary for the successful encapsulation and melanisation of parasitoid wasp eggs and larvae. Therefore, we used flow

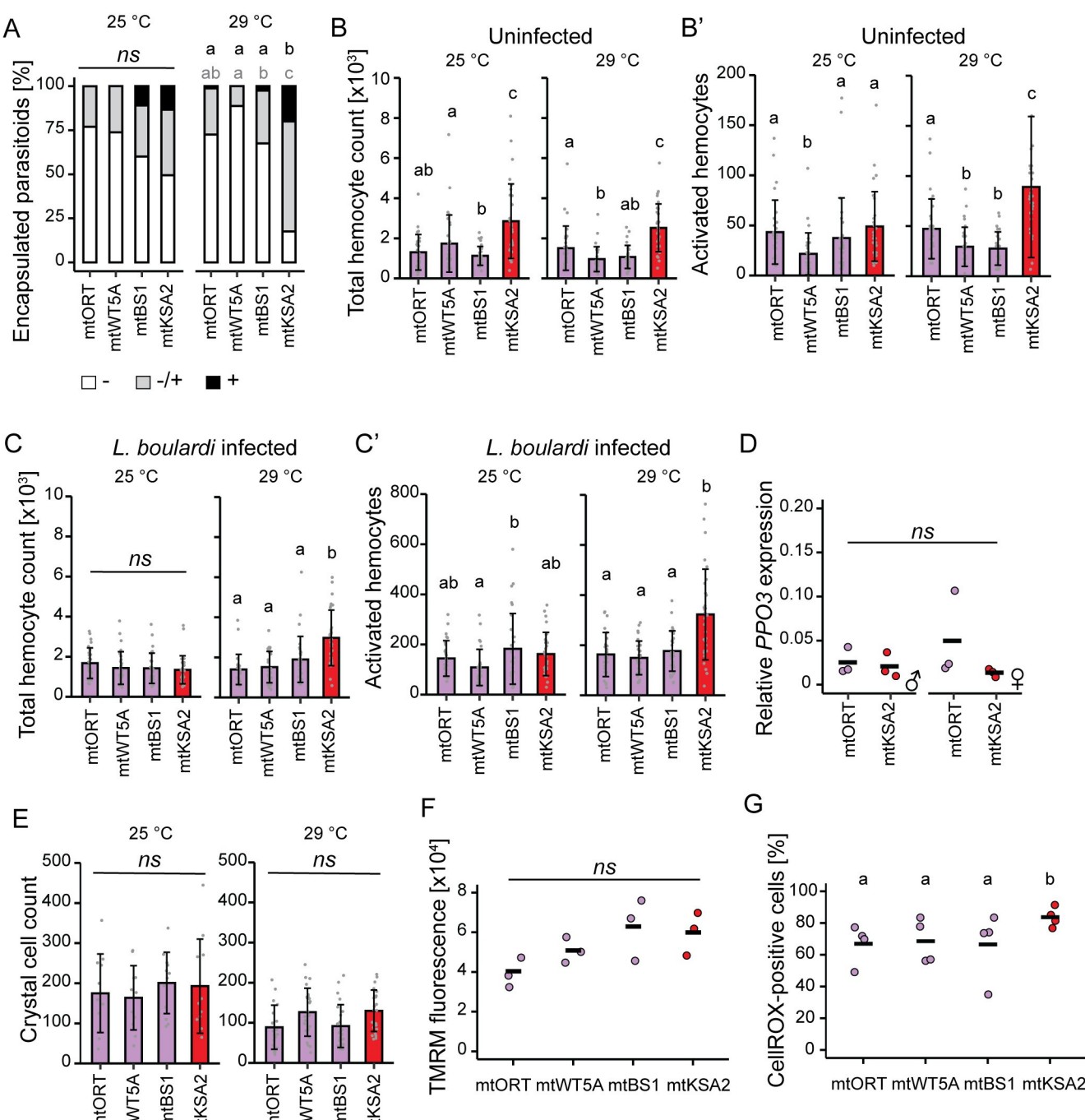

**Fig 8. mtKSA2 variation modulates the cell-mediated innate immune response in larvae.** (**A**) The effect of mtKSA2 and three other mitotypes on cell-mediated innate immunity was studied using a parasitoid wasp infection model. The success of the immune response was categorized as (-) a completely failed response with living, unmelanised wasp larva, (-/+) a partially successful response with partially melanised larva and (+) a successful response with completely melanised and encapsulated wasp larva (n = 80). The data was analysed with logistic regression with a binomial distribution and Tukey post hoc tests were applied for pairwise comparisons. Grey letters indicate statistical differences when comparing proportion of partially or fully (LM + M) melanised wasp eggs/larvae among the strains, and black letters indicate the difference when comparing fully melanised (successful immune response) wasp eggs/larvae. The same letter indicates that there was no statistically significant difference, different letter indicates a statistical difference at the level of p < 0.05. (**B-C'**) Total hemocyte numbers and numbers of cells falling into the activated hemocyte gate in uninfected (**B-B'**) and wasp-infected (**C-C'**) larvae, reared either at 25°C or 29°C using a flow cytometer (n = 30). One mtKSA2 individual had a very high cell count in B' (417 activated hemocytes) and was removed from the plot to facilitate plotting. The data on hemocyte counts was analysed using a generalized linear model with a negative binomial distribution, temperatures were analysed separately. Error bars indicate standard deviation. The same letter indicates that there was no statistically significant difference, different letters indicate a statistical difference at the level of p < 0.05. (**D**) The expression of a lamellocyte specific gene, *PPO3*, was measured in the

hemocytes of mtORT and mtKSA2 male (on the left) and female (on the right) larvae reared at 29˚C (n = 3) relative to *His3.3B* expression (2^(-dCT)). (**E**) Crystal cell counts were measured in larvae reared at 25˚C and at 29˚C. Numbers of black punctate (crystal cells) were counted from the three lowest larval segments (n = 12 for 25˚C, n = 24 for 29˚C). (**F**) Mitochondrial membrane potential was assayed as the fluorescent signal intensity of TMRM dye in hemocytes (n = 3). Data were analysed using a one-way ANOVA. (**G**) Reactive oxygen species (ROS) levels were measured in the hemocytes of non-infected larvae reared at 29˚C using the CellROX[TM] green reagent (n = 4, ~2000 hemocytes per replicate) as a percentage of hemocytes stained with CellROX[TM]. The data were analysed using a logistic regression with a binomial distribution and Tukey post hoc tests were applied for pairwise comparisons.

cytometry to assess the total number of circulating hemocytes in uninfected and *L. boulardi*-infected larvae (S7A–S7C' Fig). Hemocyte counts between the male and female larvae within a mitotype differed only on a few occasions (S7D–S7E' Fig), so we pooled the hemocyte data for subsequent analyses (Fig 8B–8C'). Uninfected mtKSA2 larvae had the highest baseline hemocyte counts at both temperatures (Fig 8B). In addition, at 29˚C mtKSA2 also had more hemocytes after parasitoid infection (Fig 8C).

To test if the differences in the hemocyte numbers might be due to the differences in hemocyte viability, we analysed the proportion of dead hemocytes (PI-positive) relative to total hemocyte counts. In general, there were no major differences in hemocyte viability, but mtKSA2 larvae had a slightly smaller proportion of PI-positive hemocytes than mtORT larvae in all conditions except for in uninfected larvae reared at 29˚C (S7F-S7G' Fig). Based on the flow cytometry FSC-area vs. FSC-height plot, mtKSA2 larvae also had more hemocytes that could be categorized as immune activated (lamellocytes) in both uninfected and wasp-infected larvae at 29˚C compared to other cybrids (Fig 8B' and 8C'). We measured the expression of a lamellocyte-specific gene *prophenoloxidase 3* (*PPO3;* [38]) from the hemocytes of uninfected mtORT and mtKSA2 larvae maintained at 29˚C to test for the presence of mature lamellocytes. However, there were no differences in the expression levels of *PPO3* (Fig 8D).

In addition to lamellocytes, crystal cells, which are the main producers of melanin upon wounding, participate in the melanisation response of parasitoid wasp eggs and larvae. We quantified the amount of crystal cells from 3rd instar larvae reared at 25˚C and 29˚C and found no differences when measured from the three lowest posterior segments from the dorsal site of the larvae (Figs 8E and S8A–S8A'). However, there were differences in the presence of crystal cells in the lymph glands: mtORT = 30%, mtWT5A = 8.3%, mtBS1 = 20.8% and mtKSA2 = 75% of the larvae (n = 12) were shown to have dark punctae in the lymph gland (S8A' Fig).

Next, we assessed the efficacy of the melanisation response at the larval stage by dissecting larvae and bleeding the hemolymph onto a glass slide, after which we measured the strength of spontaneous melanisation (S8B Fig). mtKSA2 larval hemolymph had a significantly stronger melanisation response compared to mtORT or mtWT5A, when analysed at the endpoint of the experiment (S8B' Fig). Therefore, we conclude that the measured hemocyte profiles are indicative of enhanced cell-mediated innate immunity in mtKSA2 flies at the larval and adult stages regardless of infection status.

## mtKSA2 does not affect the mitochondrial membrane potential but increases ROS in hemocytes

In a previous study, we showed that hemocyte targeted OXPHOS gene knockdown causes mitochondrial depolarization, elevated hemocyte count, and immune activation [53]. To assess the effect of the four mtDNA mitotypes on mitochondrial function in hemocytes, we measured the mitochondrial membrane potential of hemocytes in uninfected cybrid larvae (Fig 8F). There were no significant differences in the mitochondrial membrane potential among mitotypes, but on average mtORT hemocytes tended to have a lower membrane potential. In general, higher mitochondrial membrane potential is related to elevated levels of ROS [54]. We

used CellROX green probe to measure ROS from larval hemocytes reared at 29˚C. First, we used hemocytes from larvae fed with the antioxidant N-acetylcysteine (NAC; 1 mg/ml of food) as a negative control, and hemocytes from wasp-infected larvae as a positive control. As expected, the highest proportion of CellROX-positive hemocytes, and also the highest Cell-ROX-intensity, were found in the untreated wasp-infected larvae, followed by descending Cell-ROX prevalence in untreated and uninfected, NAC-fed infected and finally NAC-fed uninfected larvae (S9A–S9C Fig). We found that the mtKSA2 larvae had the highest proportion of CellROX-positive hemocytes, while the remaining three cybrid lines showed similar ROS levels (Fig 8G). Taken together, our data on larval hemocytes suggest that mtKSA2 larvae have enhanced encapsulation efficiency as a result of increased hemocyte numbers. Further, the increased levels of ROS in mtKSA2 hemocytes may aid in killing the parasite. In addition to the enhanced melanisation response on the parasitoid eggs and larvae, the melanisation response of the larval hemolymph was also the strongest in mtKSA2.

## Discussion

Mitochondria have multifaceted roles in the cell besides the generation of ATP, and production of heat [55]. One of these roles concerns mitochondrion as a hub for metabolites with important functions in the immune responses [15]. We focused on the effect of mtDNA variation on innate immune responses and found it to cause heterogeneity in infection outcomes. Among a panel of cybrid lines, mitotype mtKSA2 was shown to enhance both humoral and cellular innate immune responses in adult and larval stages, respectively. The enhanced immunocompetence of mtKSA2 was not due to preactivation of humoral immune signaling pathways, but instead due to increased larval hemocyte count and upregulation of scavenger receptors expressed in hemocytes prior to and upon parasitoid and bacterial infections. Furthermore, mtKSA2 hemocytes had increased ROS-production and stronger melanisation response, further explaining the enhanced immunocompetence.

Variation in the mtDNA has been shown to affect biochemical, cellular and physiological traits in various species [18–20,56–58]. Here, we studied the effect of natural mtDNA variation on the innate immune response using a *Drosophila* cybrid model. A panel of mitotypes in the same donor nuclear background (Oregon-RT, ORT) showed that mtDNA variation modulates infection outcomes after bacterial, viral and parasitoid infections. We focused on four distinct mitotypes ([18]; S5 Table) with either weak (mtORT, mtBS1) or strong (mtKSA2 and mtWT5A) immune response against bacterial pathogens, and selected mtKSA2 and mtORT for transcriptomic profiling. mtKSA2 enhanced the immunocompetence of the host following infection with a range of pathogens. mtKSA2 harbours a single nucleotide polymorphism (SNP) D21N in the *cytochrome b* (*mt:cyt-b*) gene of OXPHOS complex III (cIII). Based on the crystal structure model, D21N is predicted to destabilize the interaction between the *mt:cyt-b* and the nuclear encoded gene *UQCR-C1* of the active site of cIII [21]. Furthermore, Bevers et al. [59] showed that among 169 sequenced mtDNA genomes of the Drosophila Genetic Reference Panel (DGRP) strains, *mt:cyt-b* contained more nonsynonymous/missense mutations than the other genes encoded by the mtDNA. This may indicate that *mt:cyt-b* maintains higher sequence variation due to possible roles in other functions beside the OXPHOS. mtKSA2 also contains an A75T SNP in the cIV gene *Cytochrome c oxidase subunit III* (*mt:COIII)*, but the effects of this mutation are assumed to be neutral [21]. Both *mt:cyt-b* and *mt:COIII* were significantly upregulated in uninfected mtKSA2 when compared to mtORT flies (Fig 5A). mtORT bearing flies were more susceptible to infections than most of the cybrid lines (Fig 1). mtORT contains an indel in *mitochondrial small ribosomal RNA* (*mt:srRNA*, *12S*) and has a slightly shorter A+T rich non-coding region when compared to other mtDNA mitotypes, resulting in

a smaller mtDNA (S5 Table) [18]. This sequence variation and additional synonymous mutations [18] between mtORT and mtKSA2 and subsequent mitonuclear interactions may result in the observed differences in the immunocompetence of the hosts.

Although the cybrid model is widely utilized to study the effect of mtDNA variation by controlling the possible variation arising from the nuclear genome [15,22,60,61], it has some drawbacks. After ten generations of backcrossing the original nuclear background of the newly created cybrid line is replaced by 99.9% of the new donor nuclear background, leaving 0.1% of the nuclear genes unreplaced. At the level of the *Drosophila* genome (around 14 000 protein coding genes) this would leave around 14 genes unreplaced. However, previous work demonstrated that specific phenotypic traits of mtKSA2 cybrids originate from the mitochondrial genome and the OXPHOS function. Salminen et al. [18] showed that mitotype mtKSA2 led to a low mtDNA copy number when present in three different nuclear backgrounds. Furthermore, when introgressed into a "mitochondrial disease" nuclear background containing a mutation in the gene *technical knockout* (*tko25t*) encoding for the mitoribosomal protein S12 [62], mtDNA copy number was increased and the egg-to-pupa development time was significantly prolonged in comparison to other introgressed mitotypes. Ultimately, the *tko25t* + mtKSA2 combination caused synthetic lethality at the late pupal stage [21]. Surprisingly, 6% of the healthy nuclear background + mtKSA2 larvae had melanotic nodules, and this increased to 65% in *tko25t* + mtKSA2 larvae [21]. The formation of melanotic nodules was attributed to OXPHOS function, as with all tested cybrid lines, feeding the larvae with cIII inhibitor Antimycin A also led to the formation of melanotic nodules, and was lethal in the case of mtKSA2 [21]. These results indicate that harbouring mitotype mtKSA2 leads to increased sensitivity to (additional) cIII-related mitochondrial perturbations. Furthermore, mtKSA2 decreases fly lifespan [21], and the mtKSA2 flies tend to spend more time sleeping and move less than other cybrid lines [19]. As the aforementioned phenotypes can be theoretically connected to the function of mitochondria, we believe that majority of phenotypic traits observed in mtKSA2 cybrids originate from the mtDNA variation and the mitochondrial functions.

Although mtKSA2 and mtORT vary in their mtDNA sequences and show variation in several infection related phenotypes, we did not observe changes in the mtDNA copy number in flies infected with bacteria, oxygen consumption in uninfected flies, or in the mitochondrial membrane potential of hemocytes from uninfected larvae. Therefore, mtDNA variation of the mtKSA2 and mtORT mitotypes does not seem to drastically affect the measured mitochondrial parameters. However, uninfected mtKSA2 flies expressed both nuclear and mitochondrial encoded respiration related genes at higher levels than mtORT flies. After upregulation at early infection stages (8 h *p.i*), OXPHOS gene expression in mtKSA2 flies quickly returned to near pre-infection levels at 20 h *p.i* but stayed elevated in mtORT. Hence, OXPHOS gene expression might be indicative of infection progression: a high demand for OXPHOS gene expression during initiation and maintenance of the immune response and a downregulation towards the resolution phase. Consequently, proper and timely OXPHOS regulation might affect the outcome of bacterial infection. OXPHOS activity can be regulated at the protein level, e.g. via phosphorylation [63]. This type of post-translational modification of OXPHOS could explain why we did not observe differences in respiration despite the upregulated expression of OXPHOS genes in mtKSA2 flies. In addition, the method we used for measuring oxygen consumption may not be sensitive enough to detect subtle changes in respiration.

In case of DCV infection, both mtORT and mtKSA2 flies tended to have a suppressed expression of OXPHOS related genes when compared to uninfected or bacteria infected flies. Of note, mtKSA2 did not provide protection against DCV, in contrast to the other infections studied herein. This might be due to stronger negative effects of DCV on mitochondrial function, which override the protective effects caused by specific mtDNA variation. Previous work

demonstrated that DCV is associated with the downregulation of genes involved in ATPase activity [48], as well as changes in locomotor activity and metabolic rate [64,65], all of which may indicate disruption of mitochondrial function. Systemic DCV infections in *Drosophila*, like the ones performed herein, lead to infection of the smooth muscles around the crop, the food storage organ, causing nutritional stress that results from intestinal obstruction [48]. Further, DCV infected cells around the crop show abnormalities such as swollen mitochondria [48], perhaps indicating a role of the mitochondria in DCV dissemination and persistence. Multiple studies revealed that the alteration of mitochondrial function enables viruses to evade the innate immune response (reviewed in [66,67]). For example, Hepatitis C Virus (HCV), a single stranded, positive sense RNA virus, induces mitochondrial fission and mitophagy (the elimination of damaged mitochondria), promoting viral persistence [68]. SARS-CoV-2, another single stranded positive sense RNA virus, alters mitochondrial morphology [69], downregulates both mitochondrial and nuclear encoded OXPHOS gene expression as well as TCA cycle genes [70,71], but can also dysregulate the expression of mtDNA genes and cause disruption of mito-nuclear crosstalk [72].

Given that mitochondria participate in immune cell activation, trigger an innate immune response [53], initiate the inflammatory response via mtDAMPs and play a role in tissue regeneration [73], it seems likely that immune priming could also be partly modulated by mitochondrial variation. Immune priming or invertebrate immune memory has been demonstrated across a range of taxa [74]. Furthermore, we aimed at finding out whether the immune response of the mtKSA2 flies could be further enhanced by priming, as uninfected mtKSA2 flies already manifested signs of immune activation. Here, we found that in general, the survival rates improved when the flies were primed with heat inactivated *P. rettgeri*, prior to infection with live *P. rettgeri*. However, mtKSA2 flies did not respond to priming, indicating that the mtKSA2 innate immune response could be endogenously primed. Immune priming in invertebrate species can cause a significant upregulation of AMPs following a live challenge [74,75], which is usually associated with enhanced protection upon reinfection. However, based on our RNA sequencing data, uninfected mtKSA2 flies had lower AMP levels compared to mtORT and displayed similar AMP induction patterns as mtORT when infected. These results suggest that the protective effect of mtKSA2 does not originate from a more robust activation of the humoral immune signaling pathways and their downstream effector molecules.

Mitochondrial variation or dysfunction tend to be most evident in tissues with high ATP consumption rate [76,77]. All fly tissues contribute to gene expression patterns we observed in our RNA sequencing data. The main immune tissues of *Drosophila*, the fat body and the hemocytes, respond to infection in distinct ways [78]. While there are some common transcriptional changes, such as upregulation of AMP expression, most of the other expression patterns are tissue specific. The fat body responds to infection by enhanced expression of translation and OXPHOS related genes, whereas hemocytes upregulate genes related to phagocytosis, cell adhesion and cytoskeleton remodeling [78]. Our previous work showed that fat body specific knockdown of nuclear encoded OXPHOS genes delayed the development and weakened the immune response of the flies against parasitoid wasps [53]. However, hemocyte-specific knockdowns led to increased hemocyte counts, hemocyte activation and formation of melanotic nodules prior to an immune challenge, and ultimately enhanced the immune response against the parasitoid [53]. Furthermore, knock down of the cIII gene *UQCR-C1* in hemocytes caused a decrease in the mitochondrial membrane potential, inducing the mitochondrial unfolded protein response (UPR$^{mt}$), which is activated upon mitochondrial stress to maintain mitochondrial integrity [53,79].

Relatively mild mitochondrial perturbation caused by genetic or environmental stressors can activate or upregulate pathways that enhance the ability to withstand more severe stress, a

phenomenon referred to as mitohormesis [80]. Research suggests that factors potentially eliciting a hormetic response improve organismal health [80,81]. For example, elevated levels of ROS can elicit a mitohormetic response [82,83] and the promotion of immune competence is a potential outcome of mitohormesis [84,85]. Here, the host level effect of mtKSA2 on mitochondrial function seems to be milder than the one seen when *UQCR-C1* was knocked down in hemocytes [53]. We did not detect changes in the mitochondrial membrane potential, nor did we detect gene expression patterns indicative of activated UPR^mt when the transcriptomes of uninfected and infected mtKSA2 and mtORT flies were compared. mtDNA variation is present in all tissues of the cybrid fly lines. However, the whole fly RNA sequencing samples may mask tissue-specific responses against mitochondrial stress, e.g., due to different metabolic needs of the tissues, as well as their responses upon infection.

Interestingly, we detected changes in the abundance and function of the hemocytes in mtKSA2 flies when compared to other mitotypes. The expression of the phagocytic receptors was upregulated in uninfected mtKSA2 flies, indicating either increased number of immune cells or increased expression of phagocytic receptors to facilitate phagocytosis. Phagocytosis has been shown to have a role in immune defense against the intracellular pathogen *S. aureus* [86,87] and there is some evidence for its role in response to *P. rettgeri* [88]. An enhanced immune response against bacteria in mtKSA2 flies might therefore result from the increased number of immune cells and enhanced cell-mediated innate immunity. Indeed, mtKSA2 larvae had elevated hemocyte numbers when compared to the other mitotypes and they also produced more ROS. However, this was evident only when the larvae were reared at 29˚C and not when reared at 25˚C. Furthermore, mtKSA2 had a much stronger encapsulation response against wasps at 29˚C compared to the other mitotypes. Possible explanations for this observation include the fact that higher temperatures enhance metabolic rates [89] and increase ROS production [90]. The mtKSA2 line was originally collected from Southern Africa, and it is possible that the optimal temperature for the function of this mitotype is higher than for the other three mitotypes, which were collected in Europe or the United States [18]. Although mtKSA2 exhibits an increase in total hemocytes, it remains unclear if it also induces lamellocyte formation at the basal level or affects some other aspect of hemocyte immune activation that is beneficial for the cellular immune response. mtKSA2 also had more crystal cells in the lymph gland and a significantly stronger hemolymph melanisation response than mtORT and mtWT5A. Based on our results, mtKSA2 has a hemocyte profile that resembles the immune activated cell-mediated innate immunity and dampened AMP profile prior to infection, which leads to improved responses against parasitic invaders, as well as bacterial pathogens.

Our work illustrates that the *Drosophila* model offers a powerful means to disentangle variation in immune responses arising from mitochondrial variation, particularly from mtDNA polymorphisms which receive less attention in comparison to variation caused by the nuclear genome. Our results therefore contribute to the understanding of innate immunity by providing information about the genetic causes underlying the variation seen in the immune response.

## Materials and methods

### *Drosophila melanogaster* cybrid strains and rearing conditions

The *D. melanogaster* cybrid strains were created by backcrossing virgin females of nine mtDNA genome donor strains (originally from the Bloomington Drosophila Stock Center) with the nuclear genome donor males of the Oregon-RT strain (ORT, OregonR strain from the University of Tampere, Finland). Virgin females were collected from each generation and crossed with the ORT males for >10 generations [18]. Finally, the mitotype specific SNPs or

indels were sequenced from each of the cybrid lines to confirm the presence of the correct mitotype [18]. We thank Emerita Laurie S. Kaguni and Emeritus Howard T. Jacobs for gifting us the cybrid strains in 2017. Based on the mtDNA sequence variation, the ten mitotypes used in this study cluster into two haplogroups; haplogroup I mitotypes have a multicontinental origin (mtKSA2, mtPVM, mtOR, mtBOG1, mtWT5A and mtM2), whereas haplogroup II mitotypes (mtPYR2, mtLS, mtBS1 and mtVAG1) all have European origin [18].

Flies were cultured on a standard diet of cornmeal Lewis medium [91] for the bacterial and viral experiments. Larval experiments were conducted in a different laboratory, where mash potato -based food medium was used (36 g mashed potato powder, 9 g agar, 45.5 ml corn syrup, 14.5 g dry yeast, 8 g nipagin and 5 g ascorbic acid per 1 liter of water). Flies were maintained at 25°C ± 1°C in 12:12LD cycle. Some of the larval stage experiments were performed at 29°C ± 1°C, as stated in the text. Cybrid lines with the mtDNA mitotypes mtORT, mtWT5A, mtKSA2 and mtBS1 were chosen for more detailed experiments (S5 Table) and mtORT and mtKSA2 cybrids were used in RNA sequencing analysis.

## Bacterial infections

1–4-day old, mated females and males of a panel of ten cybrid lines were systemically infected with LYS-type peptidoglycan containing Gram-positive bacteria *Staphylococcus aureus* (PIG1) or DAP-type peptidoglycan containing Gram-negative bacteria *Providencia rettgeri* (DSM4542). Bacteria were grown overnight in LB growth media at 37°C and the inoculum was regrown to reach the exponential growth phase based on the spectrophotometer value prior to diluting and resuspending the sample in 1 x PBS to a final concentration of OD600 = 0.1. Flies were infected by pricking with a sharp 0.15 mm tungsten needle, dipped in the bacterial solution. To control the depth of the stabbing injury, the tip of the tungsten needle was bent 90° and the flies were stabbed with the bent part of the needle. Flies were stabbed in the mesopleuron at the thorax region to minimize injuries. To monitor the original bacterial dose, four females and four males were individually homogenized in 100 μl of 1 x PBS following pricking with the bacteria dipped needle. The total homogenates were plated on nonselective LB-agar plates and the numbers of bacterial cells (colony forming units, CFUs) were counted. The OD600 = 0.1 infection dose resulted on average to 150 bacterial cells per *P. rettgeri* infected fly and on average 30 bacterial cells in *S. aureus* infections. *P. rettgeri* and *S. aureus* infections were performed with six replicate vials of 10 flies each. Flies infected with *P. rettgeri* or *S. aureus* were monitored every 2 hours to record the mortality rates until the survival curves started to plateau.

## Bacterial load

CFUs were measured from flies 8 and 20 hours post *P. rettgeri* and *S. aureus* infection. CFUs were also measured 48 hours after *P. rettgeri* infection and 72 hours after *S. aureus* infection in live flies. Eight flies per sex/ strain/ treatment/ timepoint were kept in separate vials post infection. To kill the cutaneous bacteria, the flies were individually placed in 100 μl of 70% ethanol for 30–60 seconds. Ethanol was removed and 100 μl of 1 x PBS was added to the vials prior to homogenization with a plastic pestle. 10-fold dilution series were prepared in 96-well plates and 3 μl of each dilution was plated on a non-selective LB-agar plate. The original homogenate (1 fly homogenized in 100 μl 1 X PBS) was diluted 1:10–1:100000 for the CFU measurements, but in some cases even more diluted samples were required. Agar plates were incubated overnight at 29°C and CFUs were counted using a light microscope. Samples were collected directly after the infection as described above to monitor the original infection dose.

Uninfected flies were homogenized as negative controls, which were obtained by sampling one female and one male from vials prior to infection.

## Induction of humoral immune response prior to infection

We grew *P. rettgeri* overnight and heat killed the bacteria by heating it for 30 minutes at 80°C. Heat killed (HK) bacteria were pelleted down and diluted to OD 0.1. Both the undiluted and the OD 0.1 sample were plated to confirm the absence of live bacteria. 2–4 days old virgin females and males were exposed to one of the three treatments by pricking with a tungsten needle: 1) exposed to heat killed bacteria and 18 hours later pricked with 1 x PBS, 2) exposed to heat killed bacteria and 18 hours later exposed to live *P. rettgeri* (OD 0.1) and 3) exposed directly to live *P. rettgeri*. Survival was followed for 50 hours after the final treatment.

## Viral infections

Drosophila C virus (DCV) is a single stranded positive sense RNA virus that belongs to the *Cripavirus* genus. Systemic infection with DCV was introduced to 2–5 days old, mated females and males by pricking as described for the bacterial infections. The DCV inoculum of $10^9$ viral particles was thawed on ice and diluted with 10mM Tris-Hcl pH7.2 buffer to $10^8$ viral particles. Six replicate vials of ten flies per sex were used for DCV infections and four replicate vials of ten flies were used as a mock control, pricked with 10mM Tris-Hcl buffer. Survival was monitored once a day for 14 days. Male flies were also infected with Kallithea virus, a double stranded DNA virus. Females were excluded from the experiment as Kallithea mainly affects the survival rates of males [47]. Survival rates were monitored once a day for 22 days after microinjecting $10^8$ viral particles. 5–7 replicate vials of ten males per vial were used in the infections and 5 replicate vials of controls, microinjected with 1 x PBS.

## Sampling and total RNA extraction for RNA sequencing

1–4-day old virgin mtORT and mtKSA2 females and males were infected with *P. rettgeri*, *S. aureus* and DCV as described above. We collected four replicate vials of 10 females and males at 8 and 20 hours post bacterial infections, 3 days after exposure to DCV, and from age-matched untreated controls. Prior to total RNA extractions, the flies were snap frozen on dry ice and stored at -70°C. Frozen flies were homogenized in 120 µl of TRI reagent (MRC, Thermo Fisher Scientific) using a plastic pestle. Total RNA was extracted according to the manufacturer's instructions (MRC, Fisher Scientific). RNA pellets were dissolved in 50 µl of nuclease-free water and treated with Deoxyribonuclease I according to the manufacturer's instructions (Invitrogen). The RNA concentration and the purity of the samples were measured with Nano-Drop ND-1000 spectrophotometer (Thermo Fisher Scientific) and the samples were stored at -80°C.

## RNA sequencing and data analysis

RNA sequencing and data processing was performed by Edinburgh Genomics, the Genomics and Bioinformatics core facility at the University of Edinburgh (UK). RNA sequencing libraries were prepared using Illumina TruSeq Stranded mRNA Library Prep kits and sequenced using Illumina NovaSeq. Reads were trimmed using Cutadapt (version cutadapt-1.18-venv) and trimmed for quality at the 3' end using a quality threshold of 30 and for adaptor sequences of the TruSeq DNA kit (AGATCGGAAGAGC). After trimming, reads were required to have a minimum length of 50. *D. melanogaster* (BDGP_ens107), *S. aureus* (GCS_000013425.1), *P. rettgeri* (GCF_003204135) and DCV (NC_001834.1) references were used for mapping and

the standard GTF-format annotation for counting. Reads were aligned to the reference genome using STAR version 2.7.3a [92] specifying paired-end reads and the option—-out-SAMtype BAM Unsorted. All other parameters were left at the default parameters. Reads were assigned to features of type 'exon' in the input annotation grouped by gene_id in the reference genome using featureCounts version 1.5.1 [93]. Genes with biotype rRNA were removed prior to counting. featureCounts assigns counts on a fragment basis as opposed to individual reads so that fragment is counted where one or both of its reads are aligned and associated with the specified features. Strandedness was set to 'reverse' and specified a minimum alignment quality of 10. In addition to the counts matrix used in the downstream differential analysis, a matrix of Fragments per Kilobase of transcript per Million mapped reads (FPKM) was generated using the rpkm function of edgeR version 3.28.1 [94]. Gene lengths for the FPKM calculation were the number of bases in the exons of each gene and bases were counted once where they occurred in multiple exon annotations. Gene names and other fields were extracted from the input annotation and added to the count/ expression matrices. The raw counts table was filtered to remove genes consisting of mostly near-zero counts in different treatments and filtered on counts per million (CPM) to avoid artifacts due to library depth. If applicable, a row of the expression matrix was required to have values greater than 0.1 in at least four samples, corresponding to the smallest sample group as defined by Group once samples were removed. Reads were normalized using the weighted trimmed mean of M-values method [95], passing 'TMM' as the method to the calcNormFactors method in edgeR. Differential analysis, contrasting mtORT and mtKSA2 mitotypes or infected vs. uninfected animals, was carried out with edgeR (version 3.28.1) [94]. Fold changes were estimated as per the default behaviour of edgeR in order to avoid artifacts. Statistical assessment of differential expression was carried out with the quasi-likelihood (QL) F-test. The RNA sequencing data is available at NCBI's Gene Expression Omnibus (GEO) [96,97] data repository under the accession number GSE260986. Prior to the GO term analysis, we filtered the data to exclude non-coding genes and to include protein coding genes with False discovery rate (FDR) adjusted P-value $< 0.15$ and log2 FC $\leq$ -0.5 for downregulation and $\geq 0.5$ for upregulation.

## Respirometry

200 uninfected, 1–3 days old male flies per replicate were collected and were left to recover from the $CO_2$ anaesthesia for 24 hours. Flies were quickly knocked out on ice, transferred to a chilled mortar, and crunched 50–60 times with a ceramic pestle in 500 µl of cold homogenization buffer, containing 250 mM sucrose, 2 mM EGTA and 5mM Tris/Hcl (pH 7.4). Lysates were gently brushed through a 200 µm nylon mesh to filtrate the sample. Mortar and mesh were further rinsed with 500 µl of cold homogenization buffer. Samples were kept on ice and respirometry measurements were performed immediately after removing 100 µl for measuring the protein concentration using the BCA assay kit and manufacturer's protocol (Thermo Fisher Scientific).

Hansatech oxygen electrode and Oxygraph Plus software were used for the respirometry measurements. The oxygraph chamber was filled with 475 µl of respiration buffer, containing 120 mM KCl, 1mM EGTA, 1 mM $MgCl_2$, 0.2% BSA, 5 mM $KH_2PO_4$ and 3 mM Hepes/KOH (pH 7.2). The buffer was kept in the open chamber at 25°C with constant stirring for approximately 5 minutes until the oxygen concentration stabilized. After that, 25 µl of the lysate was added to the chamber, the chamber was closed and the oxygen concentration was measured before and after the sequential addition of OXPHOS complex I, III and IV specific substrates and inhibitors (S7 Table). The oxygen consumption rate for each complex was recorded and normalized to the sample protein content.

## mtDNA copy number

mtDNA copy number was measured 8 and 20 hours after *P. rettgeri* and *S. aureus* infection and from uninfected male flies. Total DNA was isolated from pools of five flies using TRIzol™ Reagent, following the manufacturer's protocol. Snap frozen flies were homogenized using a motorized pestle and 135 μl of Trizol reagent. DNA pellets were dissolved in 30 μl of nuclease-free water. 40 ng of total DNA was used as template in quantitative PCR using primers for the mitochondrial target gene *16S* rRNA and the nuclear target gene *RpL32* (S8 Table). Reactions were performed in StepOnePlus instrument (Applied Biosystems) using the Fast SYBR Green Master Mix (Applied Biosystems) under the manufacturer's recommended conditions: 95˚C for 20 s, followed by 40 cycles of 95˚ for 3 s and 60˚ for 30 s. Three biological and two technical replicates were used. Relative mtDNA copy number was calculated as $2*2^{\wedge}dCt$, where $dCt$ = nuclearCt - mitoCt (the threshold cycle values for the nuclear gene RpL32 and mito-chondrial gene 16S).

## H₂O₂ quantification

Hydrogen peroxide ($H_2O_2$) levels were measured from 2–5 days old uninfected males and 8 hours and 20 hours post *P. rettgeri* or *S. aureus* infection. Three biological replicates of pools of five flies were used for each treatment. Flies were homogenized with a plastic pestle in 200 μl of chilled 1 x PBS. The fly debris was pelleted by centrifuging 1 min at 12,000 x g at 4˚C. Sample supernatant was used in the $H_2O_2$ measurements and in the protein assay. Protein samples were snap frozen with liquid nitrogen and stored at -70˚C. Amplex Red Hydrogen Peroxide/ Peroxidase Assay Kit (Invitrogen, Catalog no. A22188) was used for measuring the $H_2O_2$ according to the manufacturer's protocol. Amplex Red reagent reacts with $H_2O_2$ to produce the red-fluorescent oxidation product resorufin, which has an excitation and emission maxima of approximately 571nm and 585nm, respectively. A microplate reader (VarioskanFlash, Thermo Fisher Scientific) equipped for measuring 560 nm absorbance was used for detecting the resorufin which has a 1:1 stoichiometry with $H_2O_2$. Because the reaction is continuous the fluorescence was measured at multiple time points to follow the kinetics of the reaction and 45 minutes after starting the reaction was selected for the analysis. The background fluorescence was corrected by subtracting the derived values from the no-$H_2O_2$ control. Total protein concentration was measured from each fly homogenate using the Bio-Rad Protein Assay (BioRad) and the of $H_2O_2$ was normalized to the amount of mg/ml of protein.

## Parasitoid wasp assay

The parasitoid wasp *Leptopilina boulardi* strain G486 was used as a natural model to induce the cell-mediated innate immune response in *Drosophila* larvae. Female flies were allowed to lay eggs for 24 hours at 25˚C. The eggs were then either kept at 25˚C or transferred to 29˚C for further development, depending on the experiment. 2^nd instar larvae were exposed to 12–15 female wasps for two hours at RT (22˚C), after which the wasps were removed, and the larvae placed back at their rearing temperature. The encapsulation response of the *Drosophila* larvae was checked from 3^rd instar larvae, 72 hours after the infection when the larvae were reared at 25˚C and 48 hours post infection at 29˚C, by dissecting the larvae individually in a drop of water under a stereomicroscope. The cellular immune response of the larvae was determined based on their ability to encapsulate and melanise the wasp egg/larva either as i) *no melanisation* (-): the larva failed to encapsulate the wasp egg /larva and the wasp continued its development, ii) *partially melanised* (-/+): the larva was able to elicit the cellular immune response against the wasp larva, which was partially melanised (encapsulated) and *iii) melanised* (+): the wasp larva was completely encapsulated and it was terminated as a sign of a successful cellular

immune response. Three replicate crosses per genotype were made, and larvae were dissected until 50 infected larvae per replicate had been collected (150 larvae per genotype in total).

## Larval hemocyte analysis

Late 3$^{rd}$ instar larvae were gently washed in a drop of filtered water and individually placed in 20 µl of 1% BSA in 1 x PBS on a 12-well glass plate. The hemolymph was bled out by dissecting the larvae open from one side, along the full length of the larvae. 80 µl of 1% BSA in PBS was added to the sample and a BD Accuri C6 flow cytometer was used to analyse the hemocytes. Each cybrid line was tested on three separate occasions (on different weeks) and each time 10 larvae were dissected per strain. To assess the hemocyte counts, we utilized the fact that hemocytes can be easily separated from other particles based on their location in a forward scatter–side scatter plot ([36], S1A Fig). Dead cells were excluded from the analysis by Propidium iodide (PI) staining (1µl per 100 µl sample). Plasmatocytes and crystal cells are small and round in their inactivated state [98] and can be gated using the forward scatter area vs. forward scatter height plot, where round cells form a population detected at an 45˚ angle. Lamellocytes are irregularly shaped [98], which places them outside of the 45˚ angle in the forward scatter area *vs.* forward scatter height plot. These characteristics were utilized in the flow cytometry to separate plasmatocytes and crystal cells from activated immune cells appearing in circulation after an immune challenge, per Järvelä-Stölting et al. [99]. We further confirmed this gating strategy by using *in vivo* fluorescent hemocyte reporter lines *eater-GFP* (plasmatocyte reporter; [100]) and *msnF9mo-mCherry* (lamellocyte reporter; [101]; Fig S1C-S1C'). We followed the infection scheme described in section 2.2 to assess the hemocyte populations from wasp-infected cybrid larvae at 25˚C and 29˚C. Larvae were reared at 25˚C or 29˚C prior quantification of the crystal cells by heating 10–12 3$^{rd}$ instar larvae in 100 µl of 1 x PBS at 60˚C for 10 minutes. Heating induces crystal cell melanisation, making it possible to visually quantify the number of crystal cells [102]. Crystal cells were quantified from the three most posterior segments of the larvae. Larvae were placed on a drop of glycerol under a cover slip on a glass slide for imaging.

## Total RNA extraction and RT-qPCR analysis of larval hemocytes

To extract total RNA from larval hemocytes, the hemocytes were collected from pools of 50 male or female third instar larvae, in ice cold 1X PBS. Samples were centrifuged for 7 minutes at 2500 g at 4˚C and the hemocyte pellets were stored at -80˚C. Total RNA was extracted using the Single Cell RNA Purification Kit (Norgen Biotek) including DNase I treatment (RNAse free DNase I kit, Norgen Biotek) as described in the manufacturer's protocol. The RNA quality was checked with a NanoDrop ND-1000 spectrophotometer (Thermo Fisher Scientific) and each sample was diluted to 10 ng/µl and stored at -80˚C. qRT-PCR reactions were performed using the iTaq Universal Sybr green One-step kit (Bio-Rad) with the following reaction mix per sample: 5 µl of iTaq universal SYBR Green reaction mix (2x), 0.125 µl of iScript reverse transcriptase (RT), 0.3 µl of forward and reverse primers each (300 nM each), 2 µl of template RNA (20 ng per reaction) and 2.275 µl of nuclease free H$_2$O. Each experiment included a sample without RT and without an RNA template for quality control. The reactions were run with a Bio-Rad CFX96 Real-time PCR system with a reverse transcription reaction for 10 min at 50˚C, polymerase activation and DNA denaturation for 1 min at 95˚C followed by 39 cycles of amplification (denaturation for 10 sec at 95˚C, annealing/extension for 15 sec at 60˚C) and a melt-curve analysis at 65–95˚C in 0.5˚C increments. Two technical and three biological replicates were included for each sample. Expression levels were measured from *PPO3* (*Prophenoloxidase 3*) and *Histone H3.3B* (*His3.3B*) was used as a control gene (S8 Table). Efficiency of

each primer pair was calculated based on a 5-fold serial dilution of the template RNA (each point assayed in duplicate), using the following formula: Efficiency (%) = (-1/10$^{slope}$– 1) x 100.

### ROS in larval hemocytes

Pools of ten male and ten female late third instar larvae per strain reared at 29˚C were washed three times with filtered water and then dissected in 250 μl of 1 x Schneider's Drosophila Medium (Gibco) on ice to obtain hemolymph. The samples were incubated with 500 nM of the CellROX green reagent (molecular probes for Life Technologies) diluted in Dimethyl Sulfoxide (DMSO) for 1 hour at room temperature in the dark. After incubation, hemocytes were stained with 2.5 μl of Propidium iodide (PI) and immediately analyzed using the BD Accuri C6 flow cytometer. Each cybrid line was tested four times. CellROX green was detected using a 530/33 nm filter and PI using a 610/20 nm filter, excitation was with the 488nm laser. 2000 live hemocytes were analysed per sample, each sample contained ~$10^4$ hemocytes in total. One-color controls (non-fluorescent, PI-only and CellROX-only hemocytes) were used to set the gates (S9A–S9B Fig).

### Mitochondrial membrane potential

To measure the mitochondrial membrane potential in larval hemocytes, the MitoProbe TMRM Assay Kit for Flow Cytometry (Invitrogen) was used. Three replicate samples for each cybrid line were analyzed, and each sample contained hemocytes from 30 male 3$^{rd}$ instar larvae. The larvae were washed, dissected in 1x PBS, and stained with 0.4 nM TMRM for 30 minutes in the dark. To distinguish between living and dead cells, SYTOX Blue Dead Cell Stain (Invitrogen) was used. The hemocytes were incubated with the stain (0.1 nM) for 15 minutes. The samples were analysed using the CytoFlex S flow cytometer (Beckman Coulter). The following channels and laser gain settings (in parentheses) were used: FSC (20), SSC (40), PE (for TMRM detection, 80), PB450 (for SYTOX detection, 40).

### Hemolymph melanisation assays

Cybrid flies were kept at 25˚C and vials with eggs were transferred to 29˚C for egg-to-larva development. Pools of three third instar larvae were dissected per well on a 12-well slide in a drop of 1 x PBS to release the hemolymph. Carcasses were removed and the slides were imaged as soon as possible (within 10 minutes of the dissection start) using a stereomicroscope and Nikon DS-Fi2 camera. Slides were put in a container with wet tissue paper at the bottom to prevent the hemolymph samples from drying, and imaged at 15, 30, 170 and 260 minutes after the start of dissection. Samples were left in the container overnight and imaged once more the next morning (endpoint of the experiment, ~ 24 h after dissections). We conducted the experiment on two subsequent days (2 experimental blocks) and pooled the data from these experiments. Melanisation intensity was quantified using ImageJ2 (version 2.14.0/1.54f).

### Statistical analyses

We used R versions 4.0.2.– 4.3.2. to analyze the data [103]. For plotting, ggplot2 [104] was utilized. To test for overall differences in survival, Log-Rank (Mantel-Cox) test was applied using the GraphPad Prism 9 software. For hazard ratios, log-rank tests were applied using the Survival package [105]. To test for differences in CFUs (Fig 1C-1D'), linear models with a natural log transformed response were fitted and the significance of the interaction effects and the main effect of the mitotype was assessed using Type III sums of squares, (Anova function available in the car package) [106]. Model fits were assessed using standard diagnostic plots. For

the priming data (S3 Fig), Cox mixed effects models [105,107] were fitted to test for differences in fly survival, using hour of death as the response variable, strain and sex as dependent variables and vial as a random intercept. Flies that were dead at the end of the experiment were included as censored cases. After running the survival models, we tested whether they fulfilled the assumptions of proportional hazards over time using the cox.zph function [108]. Neither model fulfilled the assumptions of proportional hazards over time, so we used the survival regression function (survreg) to compare results. In both cases, the coxph models and exponential survival regression models gave qualitatively similar results so we present the results of the Cox mixed effect models for ease of interpretation. We analyzed differences in $H_2O_2$ levels using linear models with a natural log transformed response where appropriate (Fig 6). P-values were corrected for multiple testing using the p.adjust function and a Holm correction. RT-qPCR, mtDNA copy number and respiration data were analyzed by modeling mitotype as independent variable and each replicate experiment as a random effect, followed by Tukey's HSD for pairwise comparisons. The data on melanisation of wasp eggs and larvae were analysed using a generalized linear model with binomial distribution using the lm4 package [109] with fully melanised or fully + partially melanised cases denoted as a success and non-melanised as a failure, and mitotype as an independent variable. Tukey method in the multcomp package was used for pairwise comparisons between the mitotypes [110]. The hemocyte count data were analyzed using a negative binomial generalized linear model using the MASS package [111], with mitotype as independent variable and replicate experiment as a random effect. The least square means (estimated marginal means) were analyzed for multiple comparisons using the emmeans package and the Tukey method was used for adjusting the p-value [112]. The data on the proportion of PI-positive (live) hemocytes was analysed using a generalized linear model with a binomial distribution (lme4 package) with live hemocytes as a success and dead hemocytes as a failure, mitotype as an independent variable and replicate as a random effect. For comparisons between the mitotypes, pairwise contrasts were analyzed using the emmeans package. Adobe Illustrator version 27.9 was used to edit and compile the figures. Data and code for statistical analyses are available at Zenodo (https://zenodo.org/records/14162508).

## Supporting information

**S1 Fig. Females tend to be more susceptible towards *P. rettgeri* and *S. aureus* infection, independent of the mitotype.** Sex-specific hazard ratios of survivals after (**A**) *P. rettgeri* and (**B**) *S. aureus* infection. Female to male hazard ratio is calculated based on the results from a Cox-proportional hazard model. Error bars denote 95% confidence intervals. 1 = absence of sexual dimorphism, >1 = greater susceptibility in females. Haplogroups based on [18]. Data was analysed using Log-Rank tests. *ns* not significant; * p<0.05; ** p<0.01; *** p<0.001.
(TIF)

**S2 Fig. mtKSA2 enhances survival upon bacterial infections.** Hazard ratios of survival post (**A-A'**) *P. rettgeri* and (**B-B'**) *S. aureus* infection with mtKSA2 set as the reference. 1 = absence of mtKSA2 vs. mitotype X variation, >1 = greater susceptibility in mitotype X when compared to mtKSA2. Error bars denote 95% confidence intervals. Data was analysed using Log-Rank tests. *ns* not significant; * p<0.05; ** p<0.01; *** p<0.001.
(TIF)

**S3 Fig. The effect of immune priming depends on a mitotype.** Flies were primed with heat-killed *P. rettgeri* or left naive and then infected with live *P. rettgeri* or sham infected with PBS. Survival following priming with the heat-killed pathogen and live challenge differed significantly among mitotypes (Mitotype x Treatment: $\chi^2 = 14.77$, p = 0.002) and between males and

females (Mitotype x Sex, $X2$ = 7.553, p = 0.022). Survival data were analysed using Cox Mixed Effects models.
(TIF)

**S4 Fig. mtKSA2 enhances survival from Kallithea infection and outcompetes mtORT in DCV infection.** Hazard ratios of survivals after (**A**) Kallithea and (**B-B'**) DCV infections with mtKSA2 set as a reference. 1 = absence of mtKSA2 vs. mitotype X variation, >1 = greater susceptibility in mitotype X vs. mtKSA2. Error bars denote 95% confidence intervals. Data was analysed using Log-Rank tests. *ns* not significant; * p<0.05; ** p<0.01; *** p<0.001.
(TIF)

**S5 Fig. Principal component analysis (PCA) of normalized read counts in female samples.** (**A-A"**) PCA was used to visualize the clustering of the RNA sequencing samples of uninfected and bacterial (**A-A'**) or viral (**A"**) infected mtORT and mtKSA2 females.
(TIF)

**S6 Fig. Expression patterns of genes with predicted OXPHOS function.** Genes purported to be involved with OXPHOS according to Flybase annotation (http://flybase.org; [50]) were removed from the heatmap in Fig 5 due to their unusual expression patterns. In treatments where other OXPHOS genes were upregulated, the "OXPHOS-like (L)" genes were downregulated and *vice versa*.
(TIF)

**S7 Fig. Flow cytometer gating strategy for hemocytes, sex-specific hemocyte counts and detection of PI-positive (dead) hemocytes in larvae. (A)** Hemocytes form a population separated from cellular debris based on size and shape in the hemolymph on a forward scatter (FSC-A) vs. side scatter (SSC-A) area plot. Hemocytes in the gated area were used in subsequent analyses. (**B-B'**) Dead and dying cells become permeable to Propidium iodide (PI) stain, whereas live cells remain unstained. PI was detected using a 488 nm excitation laser and a 610/20 nm emission filter, and the PI signal is shown on the y-axis. The x-axis shows the forward scatter height (FSC-H). (**B**) Non-stained hemocytes were used to determine where the non-fluorescent cells were located in the plot. An example of the separation of live and dead hemocytes based on the PI-staining is shown in (**B'**). The live cell gate was used in the subsequent analysis of the total hemocyte counts. (**C-C′**) Gating strategy to separate steady-state plasmatocytes from activated hemocytes. (**C**) *eaterGFP*-positive plasmatocytes from uninfected larvae were gated and plotted in an FSC-A vs. FSC-H plot. The plasmatocytes are found in an approximately 45˚ angle population due to their round shape. The plasmatocyte gate contains the majority of the *eaterGFP* -positive hemocytes in the sample. The other area in the plot contains all hemocytes that deviate from plasmatocytes by shape. (**C'**) *msn-mCherry*-positive lamellocytes from an infected larva. The majority of lamellocytes fall into the "activated hemocytes" gate. (**D-E'**) Sex-specific hemocyte counts in (**D-D'**) uninfected and (**E-E'**) *L. boulardi* -infected 3$^{rd}$ instar cybrid larvae, reared at 25˚C or 29˚C. Female hemocyte counts are shown in white and male hemocyte counts are shown in gray. The data were analysed using a generalized linear model with a negative binomial distribution and temperatures were analysed separately. ns, not significant; *, p < 0.05; ** p< 0.01; ***, p< 0.001. (**F-G'**) PI-permeable (dead) hemocytes detected in the samples of cybrid larvae. (**F-F'**) Proportion of dead cells in uninfected cybrid larvae reared at 25˚C and 29˚C. (**G-G'**) Proportion of dead cells in *L. boulardi* -infected cybrid larvae reared at 25˚C and 29˚C. The data were analysed using a generalized linear model with a binomial distribution, temperatures were analysed separately. The same letter indicates that there was no statistically significant difference, different letters indicate a statistical difference at

the level of $p < 0.05$. ns not significant; * $p < 0.05$; ** $p< 0.01$; *** $p< 0.001$.
(TIF)

**S8 Fig. mtKSA2 hemolymph exhibits an enhanced melanisation response.** (**A**) Crystal cells were measured from the three lowest posterior segments from the dorsal site (framed in blue) of the $3^{rd}$ instar larvae, reared at 29˚C. (**A'**) In some cases, crystal cells were also visible and clustered in the lymph glands (circled in red) of the larvae. (**B**) Example pictures of hemolymph on glass slides at the endpoint of the experiment, (~ 24 h after dissection). "X" = no sample. (**B'**) Melanisation intensities measured as mean gray values in each well (excluding the reflection of light seen in some wells). $n = 10$ for mtORT and $n = 12$ for the other cybrid lines. Statistical analysis was performed at the endpoint of the experiment using an ANOVA and Tukey's HSD for pairwise comparisons. Differences of other mitotypes to mtKSA2 marked in the plot. ns not significant; * $p < 0.05$; ** $p< 0.01$; *** $p< 0.001$.
(TIF)

**S9 Fig. CellROX green reagent staining of the hemocytes for reactive oxygen species (ROS) measurements.** (**A**) A threshold level was set for the PI intensity based on non-fluorescent hemocytes, and (**A'**) PI- positive hemocytes were excluded from the analysis. 2000 PI-negative hemocytes were analysed from each sample. (**B**) CellROX green staining resulted in two populations of hemocytes, a CellROX-positive (CellROX+) and a CellROX negative (CellROX-) population, which could not be separated from the autofluorescence of non-stained hemocytes. (**C**) Feeding the antioxidant N-acetylcysteine (NAC) to the larvae reduced the proportion of hemocytes falling into the CellROX-positive gate, both in uninfected and in *L. boulardi* -infected larvae. It also reduced the mean intensity of the CellROX green stain in the CellROX + gate.
(TIF)

**S1 Table. Log2FCs and FDR-adjusted p-values for all comparisons in male flies.**
(XLSX)

**S2 Table. Enriched GO terms in uninfected mtKSA2 vs mtORT.**
(XLSX)

**S3 Table. Log2FCs and FDR-adjusted p-values for genes involved in of Toll and Imd pathways.**
(XLSX)

**S4 Table. Enriched infection-induced GO terms in mtKSA2 and mtORT flies.**
(XLSX)

**S5 Table. mtDNA genome variation among the four cybrid lines based on *D. melanogaster* Reference sequence KT174474.1.**
(DOCX)

**S6 Table. Log2FCs and FDR-adjusted p-values for genes encoding for scavenger receptors expressed in hemocytes.**
(XLSX)

**S7 Table. A list of OXPHOS complex substrates and inhibitors used in respirometry experiment.**
(DOCX)

**S8 Table. RT-qPCR primer sequences.**
(DOCX)

## Acknowledgments

We thank Paulina Mika for helping with the Kallithea microinjections and Lauri Paulamäki for assisting with processing the RNAseq data. The *Drosophila* work was carried out at the University of Edinburgh and at the Tampere University, at the Tampere Drosophila Facility which is partly funded by Biocenter Finland. Sequencing and initial bioinformatics analysis was carried out by Edinburgh Genomics, The University of Edinburgh.

## Author Contributions

**Conceptualization:** Tiina S. Salminen, Laura Vesala, Yuliya Basikhina, Pedro F. Vale.

**Data curation:** Laura Vesala, Yuliya Basikhina, Megan Kutzer, Tilman Tietz.

**Formal analysis:** Laura Vesala, Yuliya Basikhina, Megan Kutzer, Tilman Tietz.

**Funding acquisition:** Tiina S. Salminen, Yuliya Basikhina, Pedro F. Vale.

**Investigation:** Tiina S. Salminen, Laura Vesala, Yuliya Basikhina, Tea Tuomela, Ryan Lucas, Katy Monteith, Arun Prakash, Tilman Tietz.

**Project administration:** Tiina S. Salminen, Laura Vesala, Pedro F. Vale.

**Supervision:** Tiina S. Salminen, Laura Vesala, Pedro F. Vale.

**Visualization:** Tiina S. Salminen, Laura Vesala, Yuliya Basikhina, Megan Kutzer, Tilman Tietz.

**Writing – original draft:** Tiina S. Salminen, Laura Vesala, Yuliya Basikhina.

**Writing – review & editing:** Tiina S. Salminen, Laura Vesala, Yuliya Basikhina, Megan Kutzer, Tea Tuomela, Ryan Lucas, Katy Monteith, Arun Prakash, Tilman Tietz, Pedro F. Vale.

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
