## [Decision Letter · Decision Letter 0]

17 Jul 2024

Dear Dr Salminen,

Thank you very much for submitting your Research Article entitled 'A naturally occurring mitochondrial genome variant confers broad protection from infection in Drosophila' to PLOS Genetics.

The manuscript was fully evaluated at the editorial level and by independent peer reviewers. The reviewers appreciated the attention to an important problem, but raised some substantial concerns about the current manuscript. Based on the reviews, we will not be able to accept this version of the manuscript, but we would be willing to review a much-revised version. We cannot, of course, promise publication at that time.

If you decide to revise the manuscript for further consideration at PLOS Genetics, please aim to resubmit within the next 60 days, unless it will take extra time to address the concerns of the reviewers, in which case we would appreciate an expected resubmission date by email to plosgenetics@plos.org.

To resubmit, log into your Editorial Manager account and select the option 'Revise Submission' in the 'Submissions Needing Revision' folder.

We are sorry that we cannot be more positive about your manuscript at this stage. Please do not hesitate to contact us if you have any concerns or questions.

Yours sincerely,

Ken M. Cadigan, PhD

Academic Editor

PLOS Genetics

Monica Colaiácovo

Section Editor

PLOS Genetics

Reviewer's Responses to Questions

**Comments to the Authors:**

Reviewer #1: This is a really interesting study, comprising a huge amount of work, assaying effects of different naturally occurring mtDNA haplotypes on a range of traits related to the humoral and cell-mediated innate immune response (including but not limited to: survival following bacterial infection, bacterial loads, response to viral infection, RNA sequencing of differential gene expression across all haplotypes and infection statuses, respirometry, ROS production measurement, mtDNA copy number, and an encapsulation response to parasitoid infection).

That’s a very impressive amount of work; a credit to the authors that they engaged in such an extensive study. The reporting of results are therefore very detailed, and quite dense (since there is so much to report, and synthesise).

I think the study and results are very interesting. But I have three major points below that require careful consideration.

1. Conceptual background. I feel the current version of the Introduction is a fairly poorly-developed conceptual synthesis for why mtDNA might be important in immune system regulation. The background provided does not seem to provide a compelling rationale for the study (and given the amount of work the authors have invested, I am sure there is a compelling rationale). The introductory paragraph is framed around the premise that we don’t yet understand much of the variation we see in immune system responses, and then pivots immediately into introducing the natural history / genetics of the mitochondrion. This comes across as disjointed – I think it would be better for the authors to spend more time on discussing the factors that might mediate variation in immune responses in invertebrates, and then outline whether some of this variation is genotypic: for example, do we know if immune system variation changes across strains with different nuclear genotype? This would give the rationale for then extending an argument that there might be also good reasons to suspect mtDNA involvement (but you’d need to outline these reasons, and build a rationale). I can think of at least 2 published papers that test for mtDNA-haplotype involvement in immune system function in Drosophila – it would seem important that introducing these form part of your Introduction.

An alternative approach would be to build and present evidence for mtDNA involvement in the regulation of innate immune responses (e.g. Fang et al 2016 Protein and Cell, 7:11:16), and then to finish by asking whether variation in mtDNA genotype matters in regulating covariation in immune response function.

2. Strain creation. The mtDNA strains (cybrid lines) were created by backcrossing (presumably females with the target mtDNAs to males with the isogenic nuclear background – but this is not actually explained) over 10 generations. Although this is commonly used, this is not an amazing strategy for partitioning mtDNA from nuclear genetic effects on trait expression (unless various safeguards are built into the design). Ten generations of backcrossing results in replacement of 99.9% of the original nuclear alleles associated with each mtDNA haplotype, placing the target haplotype into a new nuclear background. 0.1% of nuclear genes remain unreplaced; which if looked at the level of unreplaced genes might be 0.1% of 20,000 = around 20 unreplaced nuclear genes. If you look at the number of unreplaced base pairs, it might be ~ 18000 unreplaced base pairs. The mitochondrial genome only consists of 13 protein coding genes, so there are actually more unreplaced nuclear protein coding genes that will differ across each strain than there are mtDNA genes; and thus it becomes difficult to impossible to conclude whether effects of strains on trait expression are due to the mtDNA effects, or the effects of the unreplaced nuclear genes, or due to other environmental factors that the flies of a particular strain shared (shared environments such as vials etc). Typically these vial sharing effects / environmental effects are very strong, but the authors don’t seem to have accounted for them in their statistical analyses (see point 3). Similarly, uncless backrossing to the isogenic strain continues indefinitely, once the strains are created they will start to accumulate mutations in their nuclear backgrounds over generations, and thus start to diverge from each other in their nuclear genotype across time (creating independent replicates of each strain can help to statistically partition true mtDNA effects from effects of residual nuclear variation across the strains). As a result of the limitation in strain creation, it seems the the results presented here probably provide “some proof of concept” support for the idea that mtDNA haplotype affects immune function via “associations and correlations”, but I am not sure the authors can establish causality using this approach; since in theory all the differences across the straits could be due to nuclear genetic variation (unreplaced, or accumulated across the nuclear backgrounds, or even environmental variation across strains).

3 Statistical methods and inferential pipeline. Simple ANOVAs are used to test effects of mtDNA haplotype for most traits (and non parametrics alternatives for mtDNA copy number), and mixed models for others (but no details are provided as to the random effects included in these cases). Not enough detail is given regarding the hierarchical structure of the dataset, and therefore it is unclear whether there are random effects that should also be modelled to account for things such as assayed flies sharing the same rearing vials / rearing environments, which if left unaccounted for would result in large levels of pseudoreplication and highly anticonservative results. This sort of structure is typically incorporated via mixed models / multilevel models, which can be conducted in packages like lme4 in R. I think it is really important these things be clarified, and the data and R scripts used made public and referenced in the paper (so readers can find the data and scripts). The authors state the Tukey’s tests were calculated, but these are not scaffolded clearly to the descriptions of contrasts between mtDNA haplotypes in the main text of the Results section, or associated figures when the authors are discussing differences in trait values between haplotypes.

Transcriptomic Data: I am not an expert on how these are best analysed and presented; but it appears the authors have generated lists of differentially expressed genes, with FDR-correction, and then perhaps made a lot of contrasts between these lists for different haplotypes in different states of bacterial / virus infection. Are these contrasts made using a statistical approach (akin to Tukey’s contrasts), or just by comparing values of expression for genes across different haplotypes / states. Given the very high number of contrasts / comparisons made here, there is a very high chance of Type 1 errors of inference, an thus it seems important that there be some means of correcting for the number of tests / comparisons being made from the RNA sequencing results. These datasets are often analysed using general linear mixed models I believe, rather than the approach that seems to be used here. Also, why was a FDR of p < 0.15 used here. Is there are precedence for this – isn’t a FDR p value expected to be much lower than 0.05?

Reviewer #2: In this manuscript, the authors study how mtDNA-nuclear DNA interactions affect immune responses. The authors exposed several isogenic fly lines with identical nuclear genetic background (Oregon RT) but different mitochondrial genomes to various infections. The authors identified the mtKSA2 mtDNA haplotype as particularly resistant to infections, showing an improved survival response to bacterial and viral infections. The authors also demonstrate that the mtKSA2 fly line presents with an increased hemocyte count and an enhanced cell-mediated innate immune response at the larval stage that potentially aids in killing parasites.

This is an ambitious manuscript with clear observations regarding the immune phenotypes. However, the manuscript does not provide a clear mechanism for how the different mtDNA haplotypes affect the immune response. Nevertheless, the manuscript describes an interesting and important observation.

Besides the immunophenotyping the manuscript mainly describes data from various RNA sequencing analysis. The authors should be aware that albeit RNAseq is an important tool to understand differences in cellular responses, especially in mitochondrial biology, the proteome, and even post-translational modifications, are not always reflected in RNAseq data. This might be one of the reasons the authors do not reproduce the observed changes for mitochondrial activity or TMRM levels. Thus, despite the initial observation a mechanism connecting mtDNA haplotype, mitochondrial function, and immune response, is missing.

The authors propose that a variant in mt:cytb is responsible for the improved immune response, due to a mild CIII defect increasing hemocyte count. Ideally, mtDNA editing in mt:cytb could be applied, to isolate the individual mtDNA variant, but I accept that this is beyond the scope of this manuscript.

However, the authors might want to consider measuring CIII activity in isolated hemocytes to strengthen their hypothesis.

Comments:

1.    The authors routinely refer to mtDNA affecting a certain response (e.g. page 6, first sentence) “Furthermore, mtDNA drastically affected the survival response of the hosts after P. rettgeri infection …"), presumably refering to the specific mtDNA haplotype or mitotype. I suggest rephrasing this so that it is not confused with mtDNA copy number.

2.    Could the authors please explain in the text how they calculate the survival response. Is it based on the assumption that 100% of the uninfected lines will survive?

3.    To this reviewer, the mtORT line seems the most suitable control line for comparisons, as this line has undergone expensive natural selection, and other mtDNA haplotypes being tested against this background. Could the authors explain in the text more clearly why they chose a different approach for comparisons?

4.    It is not clear in Figures 1 A, A', B and B' what the statistical significance refer to. What comparisons are performed and what do the significance stars refer to? Are all compared to mtKSA2? Please specify in the manuscript.

5.    I recommend presenting the differences between groups in Figures 3A´-A´´ more clearly. The colours are very similar and the individual symbols too small to distinguish.

6.    In Figure 3D (qRT PCR results) the spread for some samples is very large with a small sample number. I recommend increasing the sample size to improve the data, as no conclusion can currently be made from the data. (e.g. lines mtKSA2 mtWT5A).

7.    A Western blot of your transcript candidates would also be helpful.

8.    Please explain in the manuscript why mtWT5A was included in Figure 3D.

9.    A second experiment supporting the increased ROS production in mtKSAG2 lines, such as aconitase activities, would be suitable.

10. A more sensitive way to analyse OXPHOS function would be to measure isolated OXPHOS activities spectrophotometrically. In this way, even a mild CIII defect could be identified.

11. The authors use both stars (*) and lettering (a,b etc) to indicate statistical significance. I understand there is thought put behind this, but it becomes a bit difficult to understand the different approaches. Could the authors just use one approach throughout the manuscript?

**Have all data underlying the figures and results presented in the manuscript been provided?**

Reviewer #1: Yes

Reviewer #2: Yes

PLOS authors have the option to publish the peer review history of their article (what does this mean?). If published, this will include your full peer review and any attached files.

Reviewer #1: No

Reviewer #2: No

---

## [Decision Letter · Decision Letter 1]

29 Oct 2024

Dear Dr Salminen,

We are pleased to inform you that your manuscript entitled "A naturally occurring mitochondrial genome variant confers broad protection from infection in Drosophila" has been editorially accepted for publication in PLOS Genetics. Congratulations!

Yours sincerely,

Ken M. Cadigan, PhD

Academic Editor

PLOS Genetics

Monica Colaiácovo

Section Editor

PLOS Genetics

Aimée Dudley

Editor-in-Chief

PLOS Genetics

Anne Goriely

Editor-in-Chief

PLOS Genetics

Comments from the reviewers (if applicable):  One of the editors (KMC) has reviewed the responses of the authors to Reviewer 1's comments and found that the authors did a conscientious job in addressing this reviewer's concerns.

Reviewer's Responses to Questions

**Comments to the Authors:**

Reviewer #2: This reviewer is happy with the revised manuscript.

**Have all data underlying the figures and results presented in the manuscript been provided?**

Reviewer #2: Yes

PLOS authors have the option to publish the peer review history of their article (what does this mean?). If published, this will include your full peer review and any attached files.

Reviewer #2: No

**Data Deposition**

http://datadryad.org/submit?journalID=pgenetics&manu=PGENETICS-D-24-00449R1

**Press Queries**

---

## [Editor Report · Acceptance letter]

6 Nov 2024

PGENETICS-D-24-00449R1 

A naturally occurring mitochondrial genome variant confers broad protection from infection in Drosophila 

Dear Dr Salminen, 

We are pleased to inform you that your manuscript entitled "A naturally occurring mitochondrial genome variant confers broad protection from infection in Drosophila" has been formally accepted for publication in PLOS Genetics! Your manuscript is now with our production department and you will be notified of the publication date in due course.

With kind regards,

Anita Estes

PLOS Genetics

On behalf of:
